# Nitrogen restricts future sub-arctic treeline advance in an individual-based dynamic vegetation model

Adrian Gustafson[1, 2], Paul A. Miller[1, 2], Robert G. Björk[4,5], Stefan Olin[1], Benjamin Smith[1, 3]

[1]Department of Physical Geography and Ecosystem Science, Lund University, Sölvegatan 12, 223 62 Lund, Sweden
[2]Center for Environmental and Climate Science, Lund University, Sölvegatan 37, 223 62, Lund, Sweden
[3]Hawkesbury Institute for the Environment, Western Sydney University, Penrith, NSW 2751, Australia
[4]Department of Earth Sciences, University of Gothenburg, P.O. Box 460, SE-40530 Gothenburg, Sweden
[5]Gothenburg Global Biodiversity Centre, P.O. Box 461, SE-405 30 Gothenburg, Sweden

*Correspondence to*: adrian.gustafson@nateko.lu.se

**Abstract.** Arctic environmental change induces shifts in high latitude plant community composition and stature with implications for Arctic carbon cycling and energy exchange. Two major components of change in high latitude ecosystems are the advancement of trees into tundra and the increased abundance and size of shrubs. How future changes in key climatic and environmental drivers will affect distributions of major ecosystem types is an active area of research. Dynamic Vegetation Models (DVMs) offer a way to investigate multiple and interacting drivers of vegetation distribution and ecosystem function. We employed the LPJ-GUESS tree individual-based DVM over the Torneträsk area, a subarctic landscape in northern Sweden. Using a highly resolved climate dataset to downscale CMIP5 climate data from three Global Climate Models and two 21st century future scenarios (RCP2.6 and RCP8.5) we investigated future impacts of climate change on these ecosystems. We also performed model experiments where we factorially varied drivers (climate, nitrogen deposition and [$CO_2$]) to disentangle the effects of each on ecosystem properties and functions. Our model predicted that treelines could advance by between 45 and 195 elevational meters by 2100, depending on the scenario. Temperature was a strong driver of vegetation change, with nitrogen availability identified as an important modulator of treeline advance. While increased $CO_2$ fertilisation drove productivity increases it did not result in range shifts of trees. Treeline advance was realistically simulated without any temperature dependence on growth, but biomass was overestimated. Our finding that nitrogen cycling could modulate treeline advance underlines the importance of representing plant-soil interactions in models to project future Arctic vegetation change.

**Keywords:** Ecosystem model, forest-tundra ecotone, treeline, sub-Arctic, climate change impacts, ecosystem stability, LPJ-GUESS, biogeophysical feedbacks.

## 1. Introduction

In recent decades, the Arctic has been observed to become greener (Epstein et al., 2012; Bhatt et al., 2010). Causes include an increased growth and abundance of shrubs (Myers-Smith et al., 2011; Elmendorf et al., 2012; Forbes et al., 2010), increased vegetation stature associated with a longer growing season, and poleward advance of the Arctic treeline (Bjorkman et al., 2018). Shrubs protruding through the snow and treeline advance alter surface albedo and energy exchange with potential feedback to the climate system (Chapin et al., 2005; Sturm, 2005; Serreze and Barry,

2011; Zhang et al., 2013; Zhang et al., 2018). Warming and associated changes in high latitude ecosystems have impli-
cations for carbon cycling through increased plant productivity, species shifts (Chapin et al., 2005; Zhang et al., 2014)
and increased soil organic matter (SOM) decomposition with subsequent loss of carbon to the atmosphere. Studies of
the Arctic carbon balance have shown that the region has been a weak sink in the past (Mcguire et al., 2009; Mcguire et
al., 2012; Bruhwiler et al., 2021; Virkkala et al., 2021), although uncertainty is substantial, and it is difficult to deter-
mine accurately the strength of this sink. How climate and environmental changes will affect the relative balance be-
tween the carbon uptake, i.e. photosynthesis, and release processes, i.e., autotrophic and heterotrophic respiration, will
determine whether the Arctic will be a source or a sink of carbon in the future.
Forest-tundra ecotones constitute vast transition zones where abrupt changes in ecosystem functioning occur (Hofgaard
et al., 2012). While a generally accepted theory of what drives treeline advance is currently lacking, several alternative
explanations exist. Firstly, direct effects of rising temperatures have been thoroughly discussed (e.g., Rees et al., 2020;
Hofgaard et al., 2019; Körner, 2015; Chapin, 1983). On the global scale, treelines have been found to correlate well
with a 6-7°C mean growing season ground temperature (Körner and Paulsen, 2004) and could thus be expected to fol-
low isotherm movement as temperatures rise. A global study of alpine treeline advance in response to warming since
1900 shows that 52% of treelines had advanced while the other half was stationary (47%), with only occasional in-
stances of retreat (1%) (Harsch et al., 2009). Similar patterns have been observed on the circumarctic scale, although
latitudinal treelines might be expected to shift more slowly than elevational treelines due to dispersal constraints (Rees
et al., 2020). As trees close to the treeline often show ample storage of non-structural carbohydrates (Hoch and Körner,
2012) it has been suggested that a minimum temperature requirement for wood formation, rather than productivity,
might constrain treeline position (Körner, 2003, 2015; Körner et al., 2016).
Secondly, it has been hypothesised that indirect effects of warming might be as important or more so than direct effects
(Sullivan et al., 2015; Chapin, 1983). For example, rising air and soil temperatures might induce increased mineralisa-
tion and plant availability of nitrogen in the litter layer and soil (Chapin, 1983). Increased nitrogen uptake could in turn
enhance plant productivity and growth (Dusenge et al., 2019). Increased nitrogen uptake as a consequence of increased
soil temperatures or nitrogen fertilisation have been shown to increase seedling winter survival among seedlings of
mountain birch (*Betula pubescens ssp. tortuosa*) – the main treeline species in Scandinavia (Weih and Karlsson, 1999;
Karlsson and Weih, 1996).
Thirdly, experiments exposing plants and ecosystems to elevated $CO_2$ often show increased plant productivity and bio-
mass increase, especially in trees (Ainsworth and Long, 2005). Terrestrial biosphere models generally emulate the same
response (Hickler et al., 2008; Smith et al., 2014; Piao et al., 2013). Although difficult to measure in field experiments,
treeline position seems unresponsive to increased [$CO_2$] alone (Holtmeier and Broll, 2007). Whether treelines are re-
sponsive to increased productivity through $CO_2$ fertilisation might yield insights into whether treelines are limited by
their productivity, i.e., photosynthesis, versus the ability to utilise assimilated carbon, i.e., wood formation. However,
the extent to which increased [$CO_2$] drives long-term tree and shrub encroachment and growth remains poorly studied.
For treeline migration to occur, it is not only the growth and increased stature of established trees that is important, but
also the recruitment and survival of new individuals beyond the existing treeline (Holtmeier and Broll, 2007). Seedlings

of treeline species are sometimes observed above the treeline, especially in sheltered microhabitats (Hofgaard et al., 2009; Sundqvist et al., 2008). However, these individuals often display stunted growth and can be decades old, although age declines with elevation (Hofgaard et al., 2009). The suitability of the tundra environment for trees to establish and grow taller will thus be an important factor for the rate of treeline advance (Cairns and Moen, 2004). Interspecific competition and herbivory are known to be important modulators of range shifts of trees (Cairns and Moen, 2004; Van Bogaert et al., 2011; Grau et al., 2012). For instance, the presence of shrubs has been shown to limit tree seedling growth (Weih and Karlsson, 1999; Grau et al., 2012), likely as a consequence of competition with tree seedlings for nitrogen. Comparisons of a model incorporating only bioclimatic limits to species distributions and more ecologically complex models have also suggested interspecific plant competition to be important for range shifts of trees (Epstein et al., 2007; Scherrer et al., 2020). Thus, as a fourth factor, shrub-tree interactions could be important when predicting range shifts such as changing treeline position under future climates. Rising temperatures have been suggested as the dominant driver of increased shrub growth, especially where soil moisture is not limiting (Myers-Smith et al., 2015; Myers-Smith et al., 2018). Furthermore, a changed precipitation regime, especially increased winter snowfall, might promote establishment of trees and shrubs through the insulating effects of snow cover with subsequent increases in seedling winter survival (Hallinger et al., 2010).

A narrow focus on a single, e.g., summer temperature, or a few driving variables may lead to overestimation of treeline advance in future projections (Hofgaard et al., 2019). Dynamic vegetation models (DVMs) offer a way to investigate the influence of multiple and interacting drivers on vegetation and ecosystem processes. Model predictions may be compared with observations of local treelines and ecotones to validate assumptions embedded in the models, and to interpret causality in observed dynamics and patterns. DVMs also offer a way to extrapolate observable local phenomena to broader scales, such as that of circumarctic shifts in the forest-tundra ecotone and the responsible drivers. Here, we examine a subarctic forest-tundra ecotone that has undergone spatial shifts over recent decades (Callaghan et al., 2013), previously attributed to climate warming. Adopting an individual-based DVM incorporating a detailed description of vegetation composition and stature, and nitrogen cycle dynamics, we apply the model at high spatial resolution to compare observed and predicted recent treeline dynamics, and project future vegetation change and its implications for carbon balance and biogeophysical vegetation-atmosphere feedbacks. In addition, we conduct three model experiments to separate and interpret the impact of driving factors (climate, nitrogen deposition, $[CO_2]$) on vegetation in a forest-tundra ecotone in Sweden's sub-arctic north.

## 2. Materials and Methods

### 2.1 Study site

Abisko Scientific Research station (ANS; 68°21' N, 18°49' E), situated in the mountain-fringed Abisko Valley near Lake Torneträsk in northern Sweden, has a long record of ecological and climate research. The climate record dates back to 1913 and is still ongoing. The area is situated in a rainshadow and is thus relatively dry despite its proximity to the ocean (Callaghan et al., 2013). The forests in the lower parts of the valley consist mostly of mountain birch *Betula pubescens ssp. czerepanovii* which is also dominant at the treeline. Treeline elevation in Abisko Valley ranges between

600-800 m above sea level (a.s.l.) (Callaghan et al., 2013). Other tree types in lower parts of the valley are *Sorbus aucu-*
*paria,* and *Populus tremula,* along with small populations of *Pinus sylvestris* which are assumed to be refugia species
from warmer periods during the Holocene (Berglund et al., 1996). Soils consist of glaciofluvial till and sediments. An
extensive summary of previous studies and the environment around Lake Torneträsk can be found in Callaghan et al.
109 (2013).

Our study domain covers an area of approximately 85 km$^2$ and extends from Mount Nuolja in the west to the mountain
Nissoncorru in the east (See Fig. 2). The northern part of our study domain is bounded by Lake Torneträsk. The mean
annual temperature was -0.5 ± 0.9 °C for the 30-year period 1971-2000 (Fig. 1; Table 2) with January being the coldest
month (-10.2 ± 3.5 °C) and July the warmest (11.3 ± 1.4 °C). Mean annual precipitation was 323 ± 66 mm for the same
reference period. This reference period was chosen as it is the last one in the dataset by Yang et al. (2011).
**2.2 Ecosystem model**
We used the LPJ-GUESS DVM as the main tool for our study (Smith et al., 2001; Smith et al., 2014; Miller and Smith,
2012). LPJ-GUESS is one of the most ecologically detailed models of its class, suitable for regional and global-scale
studies of climate impacts on vegetation, employing an individual- and patch-based representation of vegetation compo-
sition and structure. It simulates the dynamics of plant populations and ecosystem carbon, nitrogen, and water ex-
changes in response to external climate forcing. Biogeophysical processes (e.g. soil hydrology and evapotranspiration)
and plant physiological processes (e.g. photosynthesis, respiration, carbon allocation) are interlinked and represented
mechanistically. Canopy fluxes of carbon dioxide and water vapour are calculated by a coupled photosynthesis and sto-
matal conductance scheme based on the approach of BIOME3 (Haxeltine and Prentice, 1996). Photosynthesis is a func-
tion of air temperature, incoming shortwave or photosynthetically active radiation, [$CO_2$], and water and nutrient availa-
bility. Autotrophic respiration has three components - maintenance, growth, and leaf respiration. Tissue maintenance
respiration is dependent on soil and air temperature for root and above-ground respiration, respectively, along with a
dependency on tissue C:N stoichiometry. All assimilated carbon that is not consumed by autotrophic respiration, less a
10% flux to reproductive organs, is allocated to leaves, fine roots and, for woody PFTs, sapwood, following a set of
prescribed allometric relationships for each PFT, resulting in biomass, height and diameter growth (Sitch et al., 2003).
Consequently, an individual in the model is assumed to be carbon limited when autotrophic respiration equals or ex-
ceeds the amount of carbon assimilated by photosynthesis. Chronically negative carbon balance at the individual level
eventually results in plant death.
The model assumes the presence of seeds in all grid cells, meaning that simulated PFTs can establish once the climate is
favourable, as defined by each PFT's predefined bioclimatic limits. The competition between neighbouring plant indi-
viduals for light, water and nutrients, affecting establishment, growth, and mortality, is modelled explicitly. Competi-
tion for light and nutrients is assumed to be asymmetric, i.e., individuals with taller canopies or larger root systems will
be advantaged in the capture of resources under scarcity. Water uptake is divided equally among individuals according
to the water availability and the fraction of each PFT's roots occupying each soil layer. Individuals of the same age co-
occurring in a local neighbourhood or patch and belonging to the same plant functional type (PFT; see below) are as-
sumed identical to each other. Decomposition of plant litter and cycling of soil nutrients are represented by a CEN-

TURY-based soil biogeochemistry module, applied at patch scale (Smith et al., 2014). Biological N fixation is represented by an empirical relationship between annual evapotranspiration and nitrogen fixation (Cleveland et al., 1999). LPJ-GUESS does not currently incorporate PFT-specific nitrogen fixation, which for instance may be associated with species that form root nodules, such as *Alnu*s spp. Additional inputs of nitrogen to the system occur through nitrogen deposition or fertilisation. Nitrogen is lost from the system through leaching, gaseous emissions from soils or wildfires Smith et al. (2014).

For this study we employed LPJ-GUESS version 4.0 (Smith et al. 2014), enhanced with Arctic-specific features (Miller and Smith, 2012; Wania et al., 2009). The combined model incorporates an updated set of arctic PFTs (described below), improved soil physics and a multi-layered dynamic snow scheme, allowing for simulation of permafrost and frozen ground. The soil scheme includes 15 equally distributed soil layers constituting a total soil depth of 1.5 meters.

Vegetation in the model is represented by cohorts of individuals interacting in local communities or patches and belonging to a number of PFTs that are distinguished by growth form (tree, shrub, herbaceous), life history strategies (shade tolerant or intolerant), and phenology class (evergreen/summergreen). Herbaceous PFTs are represented as a dynamic, aggregate cover of ground layer vegetation in each patch. In this study 11 PFTs were implemented (See Table S2.1 in supplementary material for a description of included PFTs; see Table S2.2 in supplementary material for parameter values associated with each PFT). Out of these, three were tree PFTs: boreal needle-leaved evergreen trees (BNE), boreal shade-intolerant evergreen tree (BINE) and boreal shade-intolerant broad-leaved summergreen tree (IBS). Corresponding tree species present in the Torneträsk region include *Picea abies* (BNE), *Pinus sylvestris* (BINE), *Betula pubescens ssp. czerepanovii, Populus tremula* and *Sorbus aucuparia* (IBS). Following Wolf et al. (2008), shrub PFTs with different stature were implemented as follows: tall summergreen and evergreen shrubs, corresponding to *Salix spp*. (HSS) and *Juniperus communis* (HSE) and low summergreen and evergreen shrubs such as *Betula nana* (LSS) and *Empetrum nigrum* (LSE). We also included prostrate shrubs and two herbaceous PFTs.

Gridcell vegetation and biogeophysical properties are calculated by averaging over a number of replicate patches, each nominally 0.1 ha in area and subject to the same climate forcing. No assumptions are made about how the patches are distributed within a gridcell; they are a statistical sample of equally possible disturbance/demographic histories across the landscape of a gridcell. Within each patch, establishment, growth and mortality of tree or shrub cohorts comprising individuals of equal age (and dynamic size/form) are modelled annually (Smith et al., 2001; Smith et al., 2014). Establishment and mortality have both an abiotic (bioclimatic) and biotic (competition-mediated) component. Vegetation dynamics, i.e. changes in the distribution and abundance of different PFTs in grid cells over time, are an emergent outcome of the competition for resources between PFT cohorts at the patch level within an overall climate envelope determined by bioclimatic limits for establishment and survival. The bioclimatic envelope is a hard limit to vegetation distribution, intended to represent the physiological niche of a PFT. Furthermore, the climate envelope is a proxy for physiological processes such as meristem activity that may set species ranges, but also for climatic stressors such as tissue freezing. The parameters are intended to capture broader climatic properties of each gridcell. A detailed description of each bioclimatic parameter and its respective values can be found in Supplementary Table S2.2. Disturbance is accounted for by the occasional removal of all vegetation within a patch with an annual probability of 300 $yr^{-1}$, representing random events such as storms, avalanches, insect outbreaks, and wind-throw. The study used three replicate patches

within each 50 × 50m gridcell. We judged this number sufficient to obtain a stable representation of vegetation dynam-
ics given the relative area of each gridcell and replicate patches (0.1 ha). For summergreen PFTs we slightly modified
the assumption of a fixed growing degree day (GDD) requirement for establishment, using thawing degree days (TDD;
degree days with a 0 °C basis; see Table S2.2) to capture the thermal sum requirement for establishment of new individ-
uals.

**2.3 Forcing data**

The input variables used as forcing in LPJ-GUESS simulations are monthly 2m air temperature (°C), precipitation
(mm), and incoming shortwave radiation (W m$^{-2}$) as well as annual atmospheric [$CO_2$] (ppm), soil texture (mineral frac-
tions only), and nitrogen deposition (kg N ha$^{-1}$ month$^{-1}$). Monthly air temperature and shortwave radiation are interpo-
lated to a daily time-step while precipitation is randomly distributed over the month using monthly wet-days.

**2.3.1 Historic period**

A highly resolved (50 × 50m) temperature and radiation dataset using field measurements and a digital elevation model
(DEM) by Yang et al. (2011) provided climate input to the model simulations for the historic period (1913-2000). The
field measurements were conducted in the form of transects that captured mesoscale climatic variations, i.e., lapse rates.
In addition, the transects were placed to capture microclimatic effects of the nearby Lake Torneträsk and aspect effects
on radiation influx. The temperature in the lower parts of the Abisko Valley in the resulting dataset was influenced by
the lake with milder winters and less yearly variability. At higher elevation, the temperature was more variable over the
year and the local scale variations were more dependent on the different solar angles between seasons and by aspect
(Yang et al., 2011; Yang et al., 2012) (see Fig. S1.1; supplementary materials). For a full description of how this dataset
was constructed we refer to Yang et al. (2011) and Yang et al. (2012).
Monthly precipitation input was obtained from the Abisko Scientific Research Station weather records. Precipitation
was randomly distributed over each month using the number of wet-days from the CRUNCEP v.7 dataset (Wei et al.,
2014). We assumed that local differences in precipitation can be neglected for our study domain and thus the raw sta-
tion data was used as input to LPJ-GUESS for the historic period. Nitrogen deposition data for the historic and future
simulations were extracted from the gridcell including Abisko in the dataset of Lamarque et al. (2013). Nitrogen deposi-
tion was assumed to be distributed equally over the study domain.
Soil texture was extracted from the WISE soil dataset (Batjes, 2005) for the Abisko area and assumed to be uniform
across the study domain. Callaghan et al (2013) reports that the soils around the Torneträsk areas are mainly glacioflu-
vial till and sediments. Clay and silt fractions vary between 20-50% (Josefsson, 1990) with higher fractions of clay and
silt in the birch forest and a larger sand content in the heaths. In the absence of spatial information on particle size dis-
tributions, the soil was prescribed as a sandy loam soil with 43% sand and approximately equal fractions of silt and
clay.

### 2.3.2 Future simulations

Future estimates of vegetation change were simulated for one low (RCP2.6) and one high (RCP8.5) emission scenario. For each scenario, climate change projections from three global climate models (GCMs) from the CMIP5 GCM ensemble (Taylor et al., 2012) were used to investigate climate effects on vegetation dynamics. The chosen GCMs (MIROC-ESM-CHEM, HadGEM2-AO, GFDL-ESM2M) were selected to represent the largest spread, i.e., highest, lowest and near average, in modelled mean annual temperature for the reference period 2071-2100. Only models with available simulations for both RCP2.6 and RCP8.5 were used in the selection. Monthly climate data for input to LPJ-GUESS (temperature, total precipitation, and shortwave radiation) were extracted for the gridcell including Abisko for each GCM. The number of wet-days per month was assumed not to change in the future scenario simulations, so we used the 1971-2000 climatology for this period.

The historic climate dataset by Yang et al (2011) was extended into the projection period (2001-2100) using the delta change approach, as follows. For each gridcell monthly differences were calculated between the projection climate and the dataset by Yang et al. (2011) for the last 30-year reference period in our historic dataset (1971-2000). For temperature, the arithmetic difference was extracted, while for precipitation and incoming shortwave radiation relative (i.e. geometric) differences between the two datasets were extracted. The resulting monthly anomalies were then either added (temperature) to the GCM outputs, or used to multiply (precipitation, radiation) the GCM outputs from 2001-2100, for each of the climate scenarios used. Forcing data of atmospheric $[CO_2]$ for the two scenarios were obtained from the CMIP5 project.

### 2.4 Model experiments

To investigate the possible drivers of future vegetation change we performed three model experiments. The model was forced with changes to one category of input (driver) variables (climate, $[CO_2]$, nitrogen deposition) at a time for a projection period between the years 2001-2100. A full list of simulations can be found in Table S3 (supplementary materials).

A control scenario with no climate trend (and with $[CO_2]$ and nitrogen deposition held at their respective year 2000 values) was also created. We estimated the effect of the transient climate change, $[CO_2]$ or nitrogen deposition scenarios by subtracting model results for the last decade (2090-2100) in the no-trend scenario from those for the last decade (2090-2100) of the respective transient scenario. To estimate how sensitive the model was to different factors, we performed a Spearman rank correlation for each PFT in 50 m elevational bands over the forest-tundra ecotone. We chose Spearman rank over Pearson since not all correlations were linear.

### 2.4.1 Climate change

To estimate the sensitivity to climate change the same scenarios as for the future simulations (Section 2.3.2) were used while $[CO_2]$ and nitrogen deposition were held constant at their year 2000 value.

Climate anomalies without any trend were created by randomly sampling full years in the last decade (1990-2000) from
the climate station data. The climate dataset was then extended using these data. The resulting climate scenario had the
same interannual variability as the historic dataset and no trend for the years 2001-2100. This scenario was used to in-
vestigate any lag-effects on vegetation change. This scenario also provided climate input for the nitrogen and $[CO_2]$
sensitivity tests described below.

**2.4.2 $CO_2$**

For our projection simulations we used five different $[CO_2]$ scenarios from the CMIP5 project. High (RCP8.5), medium
(RCP6.0; RCP4.5) and low (RCP2.6) as well as a 'no change' emission scenarios were used.

**2.4.3 Nitrogen deposition**

Scenarios of nitrogen deposition were obtained from the Lamarque et al. (2013) dataset. Since this dataset assumes a
decrease of nitrogen deposition after year 2000 we also added four scenarios where nitrogen deposition increased with
2, 5, 7.5 and 10 times the nitrogen deposition relative to the year 2000. These four scenarios were created to isolate the
single-factor effect of nitrogen increase without any climate or $[CO_2]$ change. The resulting additional loads of nitrogen
after the year 2000 in these scenarios were 0.38, 0.97, 1.46 and 1.9 gN $m^{-2}$ $yr^{-1}$ respectively.

**2.5 Model evaluation**

We evaluated the model against available observations in the Abisko area. Measurements of ecosystem productivity
from an eddy covariance (EC) tower were obtained for six non-consecutive years (Olsson et al., 2017). Biomass and
biomass change estimates were used to evaluate simulated biomass in the birch forest (Hedenås et al., 2011). Surveys of
historic vegetation change above the treeline were obtained from Rundqvist et al. (2011). Leaf area index (LAI) and
evapotranspiration estimates were obtained from Ovhed and Holmgren (1996).
The studies by Hedenås et al. (2011) and Rundqvist et al. (2011) were used to evaluate model outputs around the obser-
vation year 2010. To compare biomass and vegetation change with these studies we extracted five year multi-model
averages for 2008-2012 from our projection simulations (section 2.3.2). These means were used to calculate modelled
change in biomass and vegetation in our historic dataset and used to compare the modelled output to the observational
data.
To determine the local rates of treeline migration several transects were defined within our study domain (Fig. S1.2;
supplementary material). These transects were chosen to represent a large spread in heterogeneity with regard to slope
and aspect in the landscape. A subsample of the selected transects were placed close to the transects used by Van Bo-
gaert et al. (2011) and used to evaluate model performance. Results from the model evaluation are summarised in Table
1 and Table S1.1.

**2.6 Determination of domains in the forest-tundra ecotone**

In our analysis we distinguished between forest, treeline and shrub tundra, defined as follows. Any gridcell containing 30% fractional projective cover or more of trees was classified as forest. This limit has been used by other studies in the area (e.g., Van Bogaert et al., 2011) to determine the birch forest boundary. The treeline was then determined by first selecting gridcells classified as forest. Any gridcell with 4 or more neighbours fulfilling the 30% cover condition criteria was classified as belonging to the forest. The perimeter of the forest was then determined through sorting out gridcells with 4 or 5 neighbours classified as forest. Gridcells with fewer or more neighbors were regarded as tundra or forest, respectively. Gridcells below the treeline were classified as forest in the analysis and gridcells above the treeline were classified as tundra.

**2.7 Presentation of results**

We present seasonal values for soil and air temperature. These are averages of the three-month periods DJF, MAM, JJA, and SON, referred to as winter, spring, summer and autumn below. For the RCPs average values are presented with the ranges of the different scenarios within each RCP given in parenthesis. We report values of both gross primary production (GPP), which we benchmark the model against, and net primary productivity (NPP) as this is of relevance for the carbon limitation discussion.

**3. Results**

**3.1 Historic vegetation shifts**

The dominant PFT in the forest and at the treeline was IBS which constituted 90% of the total LAI (Fig. 2a-3a). The only other tree PFT present in the forest was BINE, which comprised a minor fraction of total LAI. However, in the lower (warmer) parts of the landscape BINE comprised up to 20% of total LAI in a few gridcells. Forest understory was mixed but consisted mostly of tall and low evergreen shrubs and grasses. Shrub tundra vegetation above the treeline was more mixed but LSE dominated with 51% of total LAI. Grasses comprised an additional 25% of total LAI and IBS was present close to the treeline where it comprised up to 5% of LAI in some gridcells. NPP for IBS in the forest increased from 96 gC $m^{-2}$ $yr^{-1}$ to 180 gC $m^{-2}$ $yr^{-1}$ over our historic period (1913-2000). Corresponding values at the treeline did not increase but saturated at around 60 gC $m^{-2}$ $yr^{-1}$. Above the treeline, IBS showed very low NPP values (<15 gC $m^{-2}$ $yr^{-1}$), while NPP for the dominant shrub (LSE) doubled from 20 gC $m^{-2}$ $yr^{-1}$ at treeline to 40 gC $m^{-2}$ $yr^{-1}$ in the tundra.

Between the start and end of our historic (1913-2000) simulation the treeline shifted upwards 67 elevational meters on average, corresponding to a rate of 0.83 m $yr^{-1}$. However, during the 20th century both a period (1913-1940) with more rapid warming (0.8°C) and faster tree migration rate (1.23 m $yr^{-1}$) as well as a period (1940-1980) with a cooling trend (-0.3°C) and stationary treeline occurred (Fig. 5). Between 1913-2000, the lower boundary of the treeline shifted upwards 2 meters, while treeline upper boundaries shifted upwards 123 m. These shifts corresponded to rates of 0.03 and

1.54 m yr$^{-1}$, respectively. Similar rates were also found in the transects established to test how the model simulates het-
erogeneity of treeline migration (Fig. S1.2; Table S1.1; supplementary materials) where the average migration rate was
0.87 (0.54 - 1.25) m yr$^{-1}$.
During the 1913-2000 period, annual average air temperature at the simulated treeline warmed from -2.0°C to -0.8°C.
Warming occurred throughout the year but was strongest in winter and spring where temperatures increased by 3.0°C
and 1.4°C, respectively. In contrast, both summer and autumn temperatures warmed by only 0.6°C. The resulting win-
ter, spring, summer, and autumn air temperatures at the treeline in 1990-2000 were -8.7°C, 3.3°C, 8.8°C, and -0.1°C,
respectively. The warming was also reflected in annual average soil temperature increases of a similar magnitude, by
2.1°C from -0.8°C to 1.3°C. Winter soil temperature increased with 3.7°C from -5.6°C in 1913 to -1.9°C in 2000. The
warmer soil temperatures resulted in a 4.8% simulated increase in annual net nitrogen mineralisation rate in the treeline
soils over the same period. In absolute numbers, nitrogen mineralisation increased from 1.29 gN m$^{-2}$ to 1.36 gN m$^{-2}$.
Combined with an increased nitrogen deposition load from 0.06 gN m$^{-2}$ in 1913 to 0.20 gN m$^{-2}$ in 2000 and an increased
nitrogen fixation from 0.13 gN m$^{-2}$ to 0.18 gN m$^{-2}$, plant available nitrogen was simulated to increase by 15.9%. Simu-
lated permafrost with an active layer thickness of <1.5 m was present at elevations down to 560 m a.s.l. in a few
gridcells, but was always well above the treeline. More shallow permafrost (active layer thickness <1 m) was only pre-
sent in gridcells at elevations of 940 m a.s.l. and above.

## 320 3.2 Projected vegetation shifts

During the 100 year projection period (2001-2100) treelines advanced between 45 (HadGEM2-AO-RCP2.6) and 195
(GFDL-ESM2M-RCP8.5) elevational meters in the different scenarios, corresponding to rates of 0.45 and 1.95 eleva-
tional meters yr$^{-1}$. Total LAI increased in the entire ecotone in both RCP2.6 and RCP8.5 compared to the historic (1990-
2000) values (Fig. 3b-c). The increase was more pronounced in RCP8.5, which also saw a large increase in low ever-
green shrubs (LSE) at the end of the century (2090-2100). While the forest was still dominated by IBS, evergreen trees
(BNE and BINE) increased and together comprised approximately 15% of total LAI. The fraction of evergreen trees in
the forest correlated well with the degree of warming in each scenario. Forest GPP was mainly driven by tree PFTs and
increased by 50% (12% - 99%) for RCP2.6 and 177% (98% - 270%) for RCP8.5. Above the treeline, low shrubs (LSS
and LSE) contributed most to annual GPP change, which increased by 33% (-12% - 67%) and 239% (105% - 370%) in
RCP2.6 and RCP8.5, respectively. Forest NPP, wherein IBS was always dominant, increased from 200 gC m$^{-2}$ y$^{-1}$ in
year 2000 to 300 (220-375) gC m$^{-2}$ yr$^{-1}$ and 490 (380-610) gC m$^{-2}$ yr$^{-1}$ for RCP 2.6 and RCP 8.5, respectively, over the
projection period. NPP for the same period for IBS at the treeline increased slightly from 60 gC m$^{-2}$ yr$^{-1}$ to 80 (74-90)
gC m$^{-2}$ yr$^{-1}$ and 104 (80-116) gC m$^{-2}$ yr$^{-1}$ for RCP2.6 and RCP8.5. Above the treeline NPP remained low (<25 gC m$^{-2}$
yr$^{-1}$) for IBS in all scenarios and always had a lower NPP than the most productive shrub PFT (LSE). NPP for this shrub
was 49 (24-64) gC m$^{-2}$ yr$^{-1}$ and 130 (81-180) gC m$^{-2}$ yr$^{-1}$. The productivity increase translated into a biomass C increase
of the same magnitude both in the forest and above the treeline.
The average summer air temperature at the treeline between the last decade of the historic and projection periods in-
creased by 0.3°C and 6.7°C for the coldest (GFDL-ESM2M-RCP2.6) and warmest (MIROC-ESM2M-RCP8.5) GCM
scenario, respectively. The advance of the 6°C JJA soil temperature isotherm was more rapid than the treeline advance
(Fig. 4). In the two warmest scenarios (MIROC-ESM2M-RCP8.5 and HadGEM2-AO-RCP8.5) summer soil tempera-
tures exceeded 6°C in the whole study domain. Treeline elevations in these scenarios only reached 745 and 660 m a.s.l.,
respectively. Treelines advanced almost twice as fast in RCP8.5 compared to RCP2.6, 1.55 (1.10-1.96) m yr$^{-1}$ and 0.84
(0.44-1.16) m yr$^{-1}$, respectively.

### 344 3.3 Model experiments

A slight treeline advance at the end of the projection period (2090-2100) of approximately 11 elevational meters was
seen in the control simulation. As all drivers were held constant or trend-free in this simulation, this reveals a lag from
the historical period, likely resulting from smaller trees that had established in the historic period that matured during
the projection period.

### 349 3.3.1 Climate change

Treeline advance occurred in all climate change scenarios although the rate was not uniform throughout the projection
period (Fig. 5). When driven by climate change alone, migration rates were faster compared to simulations where nitro-
gen deposition and [$CO_2$] were also changed (Section 3.2). Treeline advance in climate change-only scenarios ranged
between 60 elevational meters (HadGEM2-AO-RCP2.6) and 245 elevational meters (MIROC-ESM-CHEM-RCP8.5)
over the 100 year projection period.
Tree productivity was strongly enhanced by air temperature increase over the whole study domain (Fig. 6a). Weaker
correlations between productivity and other climate factors such as precipitation and net shortwave radiation were also
seen (Fig. S1.5; S1.6; supplementary materials). Annual precipitation increased in all climate change scenarios (Table
2). In the lower parts of the valley, the increased precipitation did not result in increased soil moisture during summer as
losses through evapotranspiration driven by temperature exceeded the additional input. Spring and autumn soil moisture
increased in the forest, mainly because of earlier snowmelt and thawing ground in spring and relatively weaker evapo-
transpiration in autumn. Above the treeline, soil moisture increased as the lower temperatures and LAI did not drive
evapotranspiration as strongly as in the lower parts of the valley and the increased moisture input thus outweighed the
slightly increased evapotranspiration.
Increased tree productivity in the forest resulted in an increased LAI of 0.3-1.5 m$^2$ m$^{-2}$ (18-90%). BNE appeared in the
forest and dominated in a few gridcells. In most places BNE constituted approximately 5% of total LAI. Tall shrub
(HSE and HSS) productivity and LAI increased in the forest. This increase was negatively correlated with temperature,
i.e., increase was highest in the coolest climate change scenarios. Above the treeline, tall shrubs showed the opposite
pattern, increasing by 8-50% to finally constitute 10-36% of total LAI.
Higher soil moisture content in spring and autumn favoured trees in the whole ecotone, while forest understory suffered
from earlier onset of the growing season with subsequent flushing of the leaf and light shading from taller competitors.
Although soil moisture in summer decreased in the forest, LAI and biomass carbon of summergreen shrubs were posi-
tively correlated with soil moisture. A higher soil moisture during summers in the wetter GCM scenarios promoted
summergreen shrubs over evergreen shrubs in the whole ecotone. As an example, vegetation composition on the tundra
above the treeline differed between GFDL-ESM2M and MIROC-ESM-CHEM under RCP8.5, where the warmer GCM
showed a 52% biomass C increase in the tall evergreen shrub, HSE. The intermediate warming scenario (GFDL-
ESM2M-RCP8.5) showed a more mixed increase of biomass carbon in HSE (20%) and HSS (24%). While annual tem-
perature differed with 3.9°C between the two scenarios, average annual precipitation only differed by 6.2 mm, yielding
a much (26%) lower JJA soil moisture in the warmest scenario (MIROC-ESM-CHEM-RCP8.5) compared to the colder
(GFDL-ESM2M-RCP8.5). A relatively higher soil moisture and subsequently lower water stress allow taller plants to
establish.
Radiation correlated positively with the growth of tree PFTs, with spring and autumn radiation found to be especially
important for height and biomass increase (Fig. S1.7; supplementary materials). Increased radiation provided a competi-
tive advantage for taller trees and shrubs to shade out lower shrubs and grasses in the forest. Shrubs above the treeline
were also favoured by increased light.
Net nitrogen mineralisation at the treeline showed great variation between different climate change scenarios, ranging
from a 4% decrease in GFDL-ESM2M-RCP8.5 to a 79% increase in the strongest warming scenario (MIROC-ESM-
CHEM- RCP8.5). In absolute terms, the latter increase corresponds to an increase from 1.35 gN $m^{-2}$ $yr^{-1}$ at the end of
the historic period (1990-2000) to 2.43 g N $m^{-2}$ $yr^{-1}$ at the end of the century (2090-2100). This is comparable to the ni-
trogen load in the 7.5× increased nitrogen deposition scenario. Interestingly, despite very different plant available nitro-
gen and warming, the two scenarios displayed a similar resulting (2090-2100) treeline elevation (Fig. 5a).
Permafrost with an active layer thickness of <1.5m disappeared completely from our study domain in all scenarios ex-
cept the coldest (GFDL-ESM2M-RCP2.6) where it occurred in a few gridcells at elevations of approximately 600 m
a.s.l. However, the shallow permafrost (<1m) had disappeared also in this scenario.
**3.3.2 CO$_2$**
[CO$_2$] increase enhanced productivity increase in most PFTs (Fig. 6b). Total GPP averaged over the forest increased
between 2-10% depending on the [CO$_2$] scenario, with the largest increase in RCP8.5 and smallest in RCP2.6. The CO$_2$
fertilisation effect was not uniform within the landscape, but stronger towards the forest edge with increases from 2% to
18% from the weakest to the strongest [CO$_2$] scenario. NPP for IBS increased uniformly over the forest with 2.5-8.4%
but decreased above the treeline. Thus, the productivity of the two dominant PFTs (IBS in the forest and LSE above the
treeline) was reinforced in their respective domains. The increased productivity translated into a 1-5% increase in tree
LAI in the forest while low shrub LAI increased with 24-77%. Likewise, increase in leaf area of low shrubs was largest
on the tundra under elevated [CO$_2$], which saw a 15-40% LAI increase in the low and high [CO$_2$] scenario respectively.
Above the treeline, the productivity of grasses and low shrubs responded strongly to the CO$_2$ fertilisation with a 350%
increase in GPP for grasses and 150% increase for low shrubs. The additional litter fall produced by the increased leaf
mass did not lead to an increase in N mineralisation. However, immobilisation of nitrogen through increased uptake by
microbes increased with 2-6% between the lowest and highest [CO$_2$] scenarios, yielding a net reduction of plant availa-
ble nitrogen. Despite productivity increases, the treeline remained stationary in all [CO$_2$] scenarios (Fig. 5b).

### 3.3.3 Nitrogen deposition

Productivity of woody PFTs was in general positively correlated with nitrogen in the different nitrogen deposition scenarios. In contrast, productivity of grasses was negatively correlated (Fig. 6c) as they suffered in competition for light with the trees. Annual GPP of trees (especially IBS) was positively correlated throughout the whole ecotone, but the increase in GPP was larger towards the forest boundaries than in the lower parts the forest when nitrogen was added. Nitrogen stressed plants in the model allocate more carbon to their roots at the expense of foliar cover when they suffer a productivity reduction (Smith et al., 2014). In the two scenarios with decreasing nitrogen deposition (RCP2.6; RCP8.5) there was an overall reduction in LAI in both the tundra and the forest of 6-10%. The largest reduction was seen in tree PFTs, which have the largest biomass and consequently will have the highest nitrogen demand, followed by tall shrubs. Low shrubs and grasses did however increase their LAI in the forest when nitrogen input decreased as a result of less light competition from trees. Above the treeline, LAI of low shrubs and grass PFTs also decreased with less nitrogen input.

In all scenarios with increasing nitrogen deposition there was an advancement of the treeline in the order of 10-85 elevational meters with smallest ($2\times$ nitrogen deposition) having the smallest change in treeline elevation and vice versa for largest input ($10\times$ nitrogen deposition) (Fig. 5c). In the scenarios where nitrogen input was constant or decreasing, the treeline remained stationary.

### 4. Discussion

In our simulations, rates of treeline advance were faster under climate change-only scenarios than when all drivers were changing. This revealed nitrogen as a modulating environmental variable, as nitrogen deposition was prescribed to decrease in both the RCP2.6 and RCP8.5 scenarios. During our historic simulations, the treeline correlated well with a soil temperature isotherm close to the globally observed 6-7°C isotherm. However, in our projection period the correlation between the treeline position and the isotherm weakened, revealing a fading or potential lag of the treeline-climate equilibrium that became stronger with increased warming. Future rates of treeline advance were thus constrained by factors other than temperature in our simulations. In contrast to previous modelling studies of treeline advance (e.g., Paulsen and Körner, 2014), we include not only temperature dependence on vegetation change, but also the full nitrogen cycle and $CO_2$ fertilisation effects (Smith et al., 2014). Scenarios with increased nitrogen deposition induced treeline advance, further illustrating the modulating role played by nitrogen dynamics in our results. Rising [$CO_2$] induced higher productivity in our simulations, but these productivity enhancements alone did not lead to significant treeline advance. Furthermore, although NPP for IBS was lower at the treeline than in the forest, it was never close to zero. Such a pattern, which was seen above the treeline, indicates stagnant growth in which the carbon costs of maintaining a larger biomass cancel any productivity increase. However, enhancement of productivity in combination with an allocation shift from roots to shoots, enabled by a greater nitrogen uptake, favoured taller plants over their shorter neighbours in the competition for light within the model. For treeline advance to occur, trees need to invade the space already occupied by other vegetation. As the model assumes asymmetric competition for nutrients, newly established seedlings have a disadvantage compared to incumbent vegetation, further slowing down the modelled rate of treeline advance. Field experiments with nitrogen fertilisation have shown that mountain birch at the treeline displays enhanced growth after nitrogen

addition (Sveinbjörnsson et al., 1992). Furthermore, fertilisation with nitrogen improved birch seedling survival above
the treeline (Grau et al., 2012), and is thus likely important for establishment and growth of new individuals to form a
new treeline. Historically, treeline positions show a strong correlation with the 6-7°C isotherm (Körner and Paulsen,
2004). These records are, however, a snapshot in time and are not necessarily a strong predictor of future treeline, with
other factors (as for nitrogen in our results) potentially breaking the link to temperature. As pointed out by others
(Hofgaard et al., 2019; Van Bogaert et al., 2011), considering climate change or temperature alone in projections of
treeline advance could potentially result in overestimation of vegetation change. Our results clearly point to nitrogen
cycling as a modulating factor when predicting future Arctic vegetation shifts.
In our simulations, the treeline advanced at similar rates to those experienced during the historic period, resulting in a
displacement of 45-195 elevational meters over the 100 year projection period. Some estimates based on lake sediments
in the Torneträsk region from the Holocene thermal maximum, when summer temperatures may have been about 2.5°C
warmer than present (Kullman and Kjällgren, 2006), indicate potential treeline elevations approximately 500m above
present in the warmer climate (Kullman, 2010). Macrofossil records from lakes in the area indicate that birch was pre-
sent 300-400 meters above the current treeline (Barnekow, 1999). Furthermore, pine might have occurred approxi-
mately 100-150 meters above its present distribution (Berglund et al., 1996). IBS emerged as the dominant forest and
treeline PFT in both our historic and projection simulations, but with larger fractions of evergreen trees (BNE and
BINE) at the end of the century (2090-2100). Mountain birch, represented by IBS in our model, has historically domi-
nated treelines in the study area, even during warmer periods of the Holocene (Berglund et al., 1996), but with larger
populations of pine (BINE) and spruce (BNE) than seen at present. Both pine and spruce have been found in high eleva-
tion lake pollen sediments, and can thus be assumed to have grown in higher parts of the ecotone during warmer periods
(Kullman, 2010). Treeline advance for the historic period in our simulations is broadly consistent with observational
studies from the Abisko region (Van Bogaert et al., 2011).
Temperature was a strong driver of tree productivity and growth in the whole ecotone in our simulations. For the his-
toric period rates of treeline advance followed periods of stronger warming. However, other factors such as precipita-
tion indirectly influenced treeline advance through changes in vegetation composition and nitrogen mineralisation. This
is illustrated by the comparison of GFDL-ESM2M and MIROC-ESM-CHEM under RCP8.5, where the intermediate
warming but wetter scenario had very similar resulting treeline elevation as the warmer scenario. While simulated
treeline position was too low compared to the treeline elevation reported by Callaghan et al. (2013), the correlation with
the globally observed 6-7°C ground temperature isotherm (Körner and Paulsen, 2004) throughout the historic period
gives confidence in the model results.
IBS at the treeline had a positive carbon balance (NPP) and was thus not directly limited by its productivity in our simu-
lations. This is consistent with observations of ample carbon storage in treeline trees globally (Hoch and Körner, 2012).
The modelled treeline is thus not set by productivity directly but rather by competition, as non-tree PFTs become more
productive above the treeline. Whether the treeline is set by productivity constraints or by cold temperature limits on
wood formation and meristematic activity has been a subject of debate in the literature (Körner, 2015, 2003; Körner et
al., 2016; Fatichi et al., 2019; Pugh et al., 2016). DVMs assume NPP to be constraining for growth. On the other hand,

trees close to the treeline have been shown to have ample stored carbon (Hoch and Körner, 2012). Furthermore, enhancement of photosynthesis through added $CO_2$ does not always result in increased tree growth close to the treeline (Dawes et al., 2013), and wood formation is slow below around 5°C, leading to a hypothesis of reversed control of plant productivity and treeline position (Körner, 2015). As has also been highlighted in this study, ecological interactions as a component in the control of treeline position has been the subject of attention in some recent modelling studies (See for ex., Scherrer et al., 2020). Such studies add an extra dimension to the discussion as they do not only consider plant physiology and hard limits to species distributions but also broadly accepted ecological concepts such as realised versus fundamental niche.

The model overestimated biomass carbon in the forest but captured historic rates of biomass increase. The overestimation was more severe closer to the forest boundaries as the model showed a weaker negative correlation between biomass carbon and elevation than observed by Hedenås et al. (2011). The mean annual biomass increase in the same dataset is, although highly variable, on average 2.5 gC m$^{-2}$ yr$^{-1}$ between 1997 and 2010. As simulated GPP and LAI were within the range of observations in the area (Rundqvist et al., 2011; Ovhed and Holmgren, 1996; Olsson et al., 2017), this indicates a coupling between photosynthesis and growth in the model that is stronger than observed. Terrestrial biosphere models often overestimate biomass in high latitudes (Pugh et al., 2016; Leuzinger et al., 2013) and potentially lack processes that likely limit growth close to low temperature boundaries. Examples of such processes are carbon costs of nitrogen acquisition (Shi et al., 2016), including costs for mycorrhizal interactions (Vowles et al., 2018), and temperature limits on wood formation (Friend et al., 2019). However, data on carbon allocation and its temperature dependence are scarce (Fatichi et al., 2019). Additionally, the overestimation in our study can be partly attributed to lack of herbivory in the model. Outbreaks of the moth *Epirrita autumnata* are known to limit productivity and reduce biomass of mountain birch in the area in certain years (Olsson et al., 2017), however, this would not fully explain the overestimation of biomass at treeline in our simulations. Since growth and biomass increment in the model do not include a direct temperature dependence, nor any decoupling of growth and productivity, we do not regard these mechanisms as necessary to accurately predict treeline dynamics. However, they might be important to accurately predict forest biomass at treeline.

To examine variability in the simulated treeline dynamics across the study area, we established a number of transects close to observation points in the landscape. Average treeline advance in the transects showed a somewhat faster and more homogenous migration than reported (Van Bogaert et al., 2011). The model does not include historic anthropogenic disturbances, topographic barriers, or insect herbivory, all of which have been invoked to explain heterogeneity of treeline advance rates and placement in the landscape (Van Bogaert et al., 2011; Emanuelsson, 1987). Furthermore, our model does not include any wind related processes such as wind mediated snow transport or compaction. Thus, our simulations result in a homogenous snowpack during the winter months with no differentiation in sheltering or frost damage that may result from different snow and ice properties. Sheltered locations in the landscape are known to promote survival of tree saplings (Sundqvist et al., 2008). For nitrogen cycling this may also mean that suggested snow-shrub feedbacks (Sturm et al., 2001; Sturm, 2005) are not possible to capture with the current version of our model. While overall rates of treeline migration were captured, local variations arising from physical barriers such as steep slopes, stony patches or anthropogenic disturbances were not possible to capture as these processes are not implemented in the model. High-resolution, local observations of vertically-resolved soil texture and soil organic matter content (see, e.g.

Hengl et al. (2017) for an example compiled using machine-learning) have the potential to improve the spatial variabil-
ity of modelled soil temperatures and nutrient cycling in our study domain.
A longer growing season favoured tree PFTs in the whole ecotone, which escaped early-season desiccation due to
milder winters and earlier spring thaw. Permafrost was only present at the highest elevations during the historic simula-
tion but had disappeared from the landscape by 2100 for all except the coolest scenario (GFDL-ESM2M-RCP2.6). The
simulated permafrost was however always well above the treeline and did not have a significant impact on the treeline
advancement. While some aspects of ground freezing are accounted for in the model, soil vertical and horizontal move-
ment caused by frost, and amelioration of such effects in the warmer future climate, are not. Such processes could affect
survival and competition among the plant functional types, especially in the seedling stage when plants are most vulner-
able to mechanical disturbance (Holtmeier and Broll, 2007). These effects could be relevant to treeline dynamics at the
high grid resolution of our study but are not included in our model.
Higher summer soil moisture in the wetter climate scenarios shifted the ratio of summergreen to evergreen shrubs in
favour of the summergreen shrubs, in line with observations (Elmendorf et al., 2012). Conversely, drier scenarios
yielded an increased abundance of evergreen shrubs, similar to what has been observed in drier parts of the tundra heath
in the Abisko region (Scharn et al., 2021). Within RCP8.5, the warmest (MIROC-ESM-CHEM-RCP8.5) and coldest
(GFDL-ESM2M-RCP8.5) scenario gave rise to very similar treeline positions at the end of the projection period (2090-
2100). The cooler scenario led to both higher soil moisture and a greater abundance of summergreen shrubs. Higher soil
moisture promoted carbon allocation to the canopy, and thus favoured the taller IBS tree PFT over tall shrubs (HSS).
Increased shrub abundance and nutrient cycling have been shown to have potentially non-linear effects on shrub growth
and ecosystem carbon cycling (Buckeridge et al., 2009; Hicks et al., 2019), and some observations indicate that changes
in the ratio of summergreen to evergreen shrubs, or an increased abundance of trees, might impact soil carbon loss
(Parker et al., 2018; Clemmensen et al., 2021). Thus, our results indicate that any future change in soil moisture condi-
tions could play an important role in the competitive balance between shrubs and trees and for carbon balance.
LPJ-GUESS assumes the presence of seeds in all gridcells and PFTs may establish when the 20-year (running) average
climate is within PFT-specific bioclimatic limits for establishment. This assumption may overlook potential constraints
on plant migration rates such as seed dispersal and reproduction. On larger spatial scales, it is likely that lags in range
shifts would arise from these additional constraints (Rees et al., 2020; Brown et al., 2018). Models that account for dis-
persal limitations generally predict slower latitudinal tree migration than models driven solely by climate (Epstein et al.,
2007). However, on smaller spatial scales, the same models predict competitive interactions to be more dominant in
determining species migration rates (Scherrer et al., 2020), and this is included in our model. In a seed transplant study
from the Swiss alps, seed viability could not be shown to decline towards the range limits of eight European broad-
leaved tree species (Kollas et al., 2012; Körner et al., 2016). Similarly, gene flow above the treeline could not be shown
to be limited to near-treeline trees in the Abisko region (Truong et al., 2007). Furthermore, tree saplings have been re-
ported to be common up to 100m above the present treeline (Sundqvist et al., 2008; Hofgaard et al., 2009). As environ-
mental conditions improve, these individuals may form the new treeline.

Above the treeline low evergreen shrubs (LSE) dominated the vegetation in both our historic and projection simulations. The productivity of shrubs and grasses was greatly enhanced by $CO_2$ fertilisation in our [$CO_2$] model experiment, and a large proportion of tundra productivity increases in our projection simulations could be attributed to rising [$CO_2$]. Physiological effects of elevated $CO_2$ on Arctic and alpine tundra productivity and growth are understudied. Free Air $CO_2$ Enrichment (FACE) experiments are generally considered the best method for quantifying long-term ecosystem effects of elevated $CO_2$ but are extremely costly and very few have been deployed in near-treeline locations. A majority of FACE experiments have been implemented in temperate forests and grasslands, yielding limited evidence of relevance to boreal and tundra ecosystems (Hickler et al., 2008). One FACE experiment situated in a forest-tundra ecotone in the Swiss Alps showed differing responses to elevated $CO_2$ among shrub species where *Vaccinium myrtillys* showed 11% increased shoot growth while *Empetrum nigrum* was unresponsive and the response of *V. gaultherioides* depended on the forest type in which it was growing (Dawes et al., 2013). Our model results indicated that shrubs are carbon limited and shrub productivity and growth consequently are responsive to $CO_2$ fertilisation.

## 5. Conclusions

In this study we examined treeline dynamics in the subarctic north of Sweden using an individual-based dynamic vegetation model at high spatial resolution. The model identified nitrogen cycling and availability as important modulating factors for treeline advance in a warming future climate. Internal cycling of nitrogen in soils provides the main source of this usually limiting nutrient for Arctic plants (Chapin, 1983). The model performed well regarding rates of shrub increase and treeline advance but overestimated biomass carbon in the treeline forest. Treeline migration rates were realistically simulated even though the model did not represent temperature limitations on tree growth. While a decoupling between productivity and growth in the model could potentially have improved estimates of biomass carbon, it was not needed to correctly predict treeline elevation. Instead, our results point to the importance of indirect effects of rising temperatures on tree range shifts, especially with regard to nutrient cycling and competition between trees and shrubs. Furthermore, soil moisture strongly influenced vegetation composition within the model with implications for treeline advance. Improving how models represent nutrient uptake and cycling, and incorporating empirical understanding of processes that determine tree and shrub growth, will be key to better predictions of Arctic vegetation change and carbon and nitrogen cycling. Models are a valuable aid in judging the relevance of these processes for subarctic treeline ecosystems.

## 6. Author contributions

AG designed the experiments with contributions from PM and SO. AG also performed necessary model code developments and carried out model simulations and data analysis. RGB and BS contributed scientific advice and input throughout the study and contributed to the writing. AG prepared the manuscript with contributions from all co-authors.

## 7. Competing interests

The authors declare that they have no conflict of interest.

**8. Acknowledgements**

We would like to thank Professor Christian Körner and one anonymous reviewer for their thoughtful and constructive reviews which greatly improved the manuscript and widened the scope of our analysis. We acknowledge the Lund University Strategic Research Areas BECC and MERGE for their financial support. Abisko Scientific Research Station generously shared the data used in preparation of the future climate projections. This research was partly funded (Paul A. Miller, Robert G. Björk) by the project BioDiv-Support through the 2017-2018 Belmont Forum and BiodivERsA joint call for research proposals, under the BiodivScen ERA-Net COFUND programme, and with the funding organisations AKA (Academy of Finland contract no 326328), ANR (ANR-18-EBI4-0007), BMBF (KFZ: 01LC1810A), FORMAS (contract no:s 2018-02434, 2018-02436, 2018-02437, 2018-02438) and MICINN (through APCIN: PCI2018-093149).

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

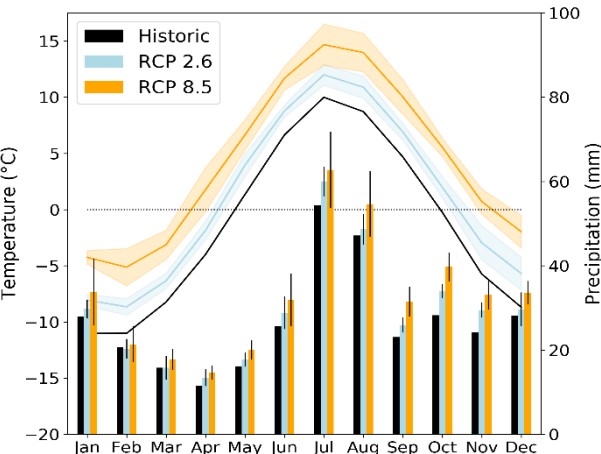

**Figure 1**. Historic (1971-2000) and projected (2071-2100) temperature (left) and precipitation (right) variability at the Abisko study area. The shaded areas (temperature) and black bars (precipitation) mark ±1 standard deviation uncertainty in the three CMIP5 multi-model mean for RCP2.6 and RCP8.5 respectively.

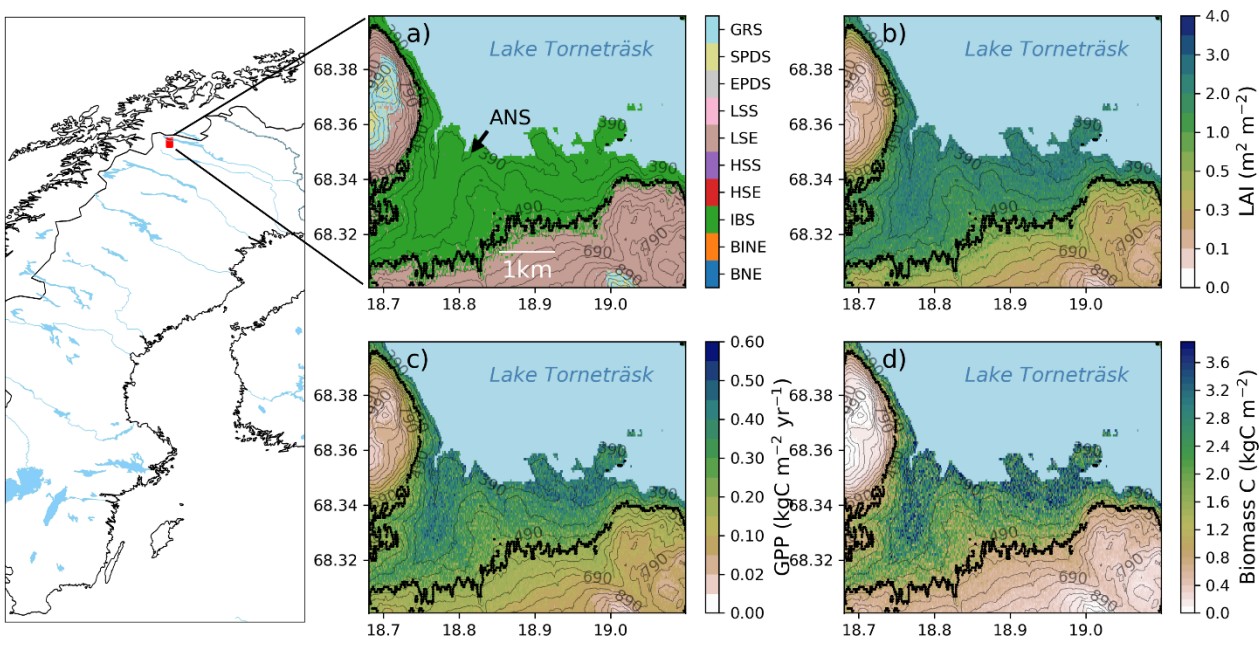

**Figure 2.** Map of Sweden and Scandinavia with a red square marking the study area. The location of the Abisko Scientific Research Station (ANS) is marked in panel a). Panels on the right show the study area in more detail and the modelled forest-tundra ecotone for the historic period (1990-2000). a) Dominant PFT (BNE – Boreal needle leaved evergreen tree; BINE – Boreal shade-intolerant needle leaved tree; IBS – Boreal shade-intolerant broadleaved tree; HSE – Tall evergreen shrub; HSS – Tall summergreen shrub; LSE – Low evergreen shrub; LSS – Low summergreen shrub; EPDS – Evergreen prostrate dwarf shrub; SPDS – Summergreen prostrate dwarf shrub; GRS - grasses) in the ecotone and total ecosystem b) LAI ($m^2$ $m^{-2}$)  c) productivity (GPP; kgC $m^{-2}$ $yr^{-1}$) and d) plant biomass carbon density (kgC $m^{-2}$). The black line in panels a-d shows the modelled treeline position. Numbers on the contour lines mark elevation in meters above sea level. Data source for map: Natural Earth

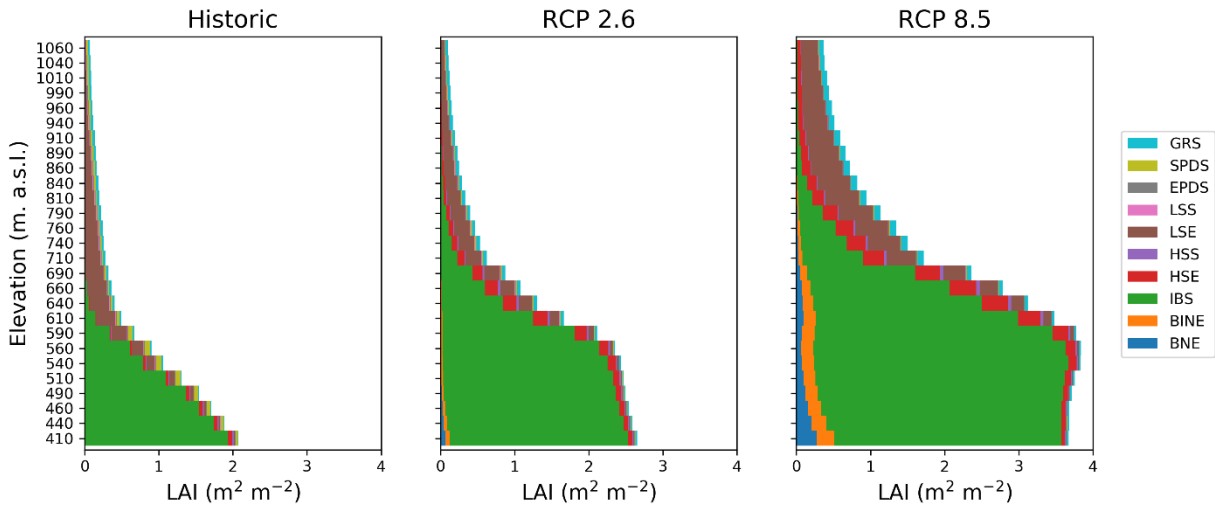

**Figure 3**. Leaf area index (LAI) in the forest-tundra ecotone for a) historic (1990-2000) and at the end of the century (2090-2100) for b) RCP2.6 and c) RCP8.5 respectively. Each bar represents 50 elevational meters.

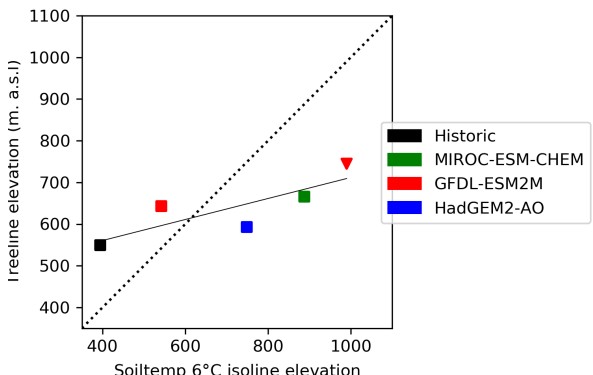

**Figure 4.** JJA 6°C soil temperature isotherm elevation relative to average treeline elevation. Square markers represent RCP2.6 while triangles represent RCP8.5. In the two warmest scenarios (HadGEM2-AO-RCP8.5 and MIROC-ESM-CHEM-RCP8.5), the 6°C soil temperatures exceed 6°C in the whole landscape. The dotted line represents the 1:1 relationship between treeline and isotherm placement while the solid line displays the treeline-soil temperature regression.

**Table 1.** Model evaluation and benchmarking results.

| Parameter | Unit | Domain | Time Interval | Model value | Observed estimate | Reference |
|---|---|---|---|---|---|---|
| GPP (Average) | gC m$^{-2}$ yr$^{-1}$ | Birch forest | 2007-2014 | 410 ± 64 | 440 ± 54 | Olsson et al., 2017 |
| Carbon density | tC ha$^{-1}$ | Birch forest | 2010 | 21.8 ± 10 | 4.39 ± 3.46 | Hedenås et al., 2011 |
| Carbon density change | % | Birch forest | 1997-2010 | 25 | 19 | |
| LAI | m$^2$ m$^{-2}$ | Forest canopy | 1988-1989 | 1.65 ± 0.66 | ~2.0 | Ovhed & Holmgren, 1996 |
| | | Understory | | 0.17 ± 0.12 | ~0.5 | |
| Densification | % | Shrub tundra | 1976-2010 | +87 ± 15 | +50-80 | Rundqvist et al., 2011 |
| Treeline elevation (min) | m. a.s.l. | Treeline | 2010 | 444 | ~600 | Callaghan et al., 2013 |
| Treeline elevation (mean) | | | | 564 | - | |
| Treeline elevation (max) | | | | 723 | ~800 | |
| Treeline elevation change (mean) | Elevational meters | Treeline | 1912-2009 | 80 | 24 | van Boogart et al., 2011 |
| Treeline elevation change (max) | | | | 123 | 145 | |
| Treeline migration rate (mean) | m yr$^{-1}$ | Treeline | 1912-2009 | +0.85 | +0.6 | van Boogart et al., 2011 |
| Treeline migration rate (max) | | | | +1.18 | +1.1 | |

**Table 2.** Seasonal temperature and precipitation for historic and scenario simulations.

| | | 1971-2000 | 2071-2100 | | | | | |
|---|---|---|---|---|---|---|---|---|
| | | Yang et al., 2011 | GFDL-ESM2M | | HadGEM2-AO | | MIROC-ESM-CHEM | |
| | Season | Historic | RCP2.6 | RCP8.5 | RCP2.6 | RCP8.5 | RCP2.6 | RCP8.5 |
| Temperature (°C) | Winter (DJF) | -9.8 | -8.2 | -5.4 | -8.1 | -4.4 | -7.4 | -3.1 |
| | Spring (MAM | -2.1 | -1.3 | 1.0 | 0.4 | 4.11 | 0.7 | 4.8 |
| | Summer (JJA) | 9.9 | 10.9 | 13.2 | 11.9 | 14.4 | 13.1 | 13.4 |
| | Autumn (SON) | 0.1 | 1.1 | 4.2 | 2.3 | 9.1 | 3.2 | 7.2 |
| | Annual (mean) | -0.5 | 0.6 | 3.3 | 1.6 | 5.0 | 2.4 | 6.6 |
| Precipitation (MM) | Winter (DJF) | 75 | 80 | 85 | 75 | 80 | 70 | 95 |
| | Spring (MAM | 45 | 40 | 45 | 40 | 45 | 50 | 55 |
| | Summer (JJA) | 125 | 130 | 130 | 130 | 150 | 135 | 145 |
| | Autumn (SON) | 75 | 90 | 95 | 85 | 95 | 95 | 110 |
| | Annual (sum) | 325 | 340 | 355 | 335 | 370 | 350 | 405 |

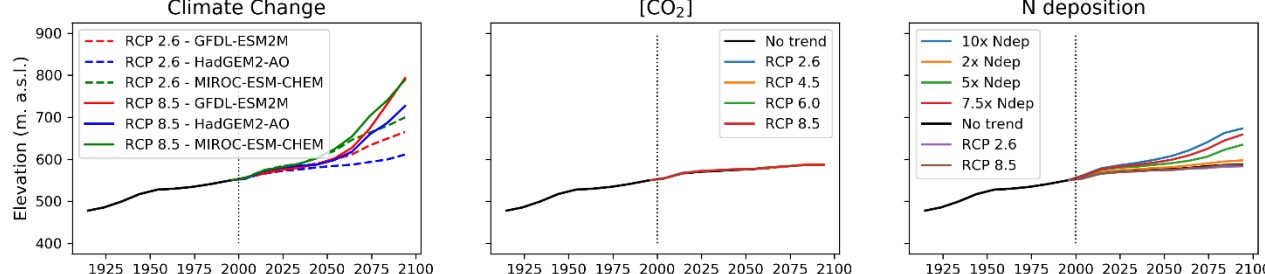

**Figure 5**. Shifts in average treeline elevation over the simulation period for the three experiments a) climate change b) $CO_2$ fertilisation and c) nitrogen deposition. Start of projection simulations are marked with a vertical dotted line in all panels. No-trend scenario in panel b-c represent a scenario where climate, $CO_2$ and nitrogen deposition are kept constant (without trend) relative to year 2000. Black line before year 2000 represents the historic simulation.

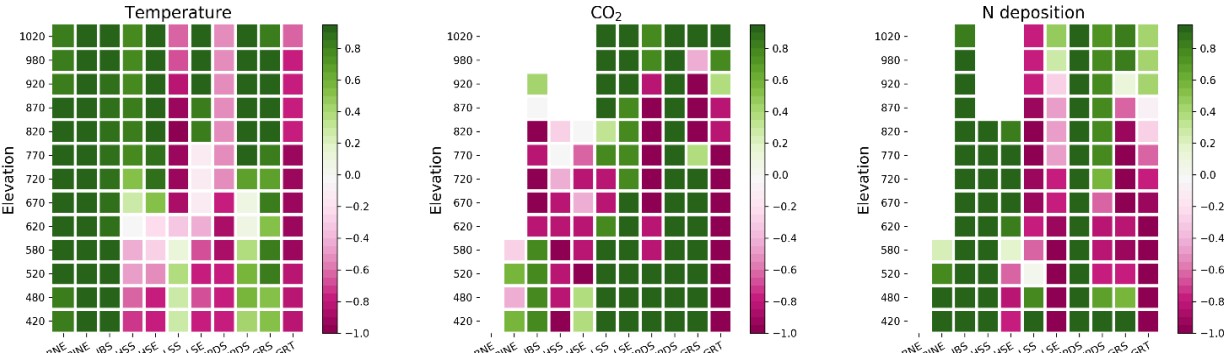

**Figure 6**. Correlation (Spearman rank) between annual GPP for each PFT and a) average 2090-2100 temperature anomalies in the climate change experiment, b) $CO_2$ scenario and c) nitrogen deposition scenario. Each box represent a 50 elevational meter band for a given PFT.