# Peer review of "Nitrogen restricts future treeline advance in the sub-arctic"

_Biogeosciences, 2021_

## Author Response (AR1)

Dear Editor,

Thank you for considering our manuscript for publication in Biogeosciences and for facilitating the review process. We are encouraged by your kind words over the manuscript. We would also like to thank the two reviewers for taking the time to scrutinise our manuscript and provide thoughtful comments. Below follows a compilation of our point-to-point answers to the reviewers' questions and concerns. The main changes consist of a broadened discussion, clarified methods and a more detailed model description. We have added discussion points on permafrost and winter processes, dispersal limitation, heterogeneity of treeline advance and the carbon limitation hypothesis. We have also added results for permafrost, N mineralisation, net primary production (NPP) and clarified the discussion where relevant for the carbon limitation hypothesis. To be consistent with how carbon fluxes are presented we have updated the units of all nitrogen related fluxes and rates, such as mineralisation and deposition, from $kgN\ ha^{-1}\ yr^{-1}$ to $gN\ m^{-2}\ yr^{-1}$. In response to points suggested by Reviewer 1 we have also improved the presentation on a few figures in both the manuscript and supplementary materials. The result is a much improved manuscript which we hope will be suitable for publication.

**Reply to reviewer 1**

*This manuscript describes a modeling study to understand the climatic and biogeochemical controls on vegetation change in sub-arctic Sweden. The authors parameterized the LPJ-GUESS dynamic vegetation model with the principle plant types found in their study area around Abisko, Sweden. Using a very high resolution local gridded climatology and downscaled GCM output, the authors then ran LPJ-GUESS in a historical and series of future climate scenarios, and in a range of sensitivity tests controlling for different processes including $CO_2$ fertilization and nitrogen cycling. The authors conclude that while climate has an overarching control on vegetation composition and position of the treeline, nitrogen availability exerts a very important influence on vegetation dynamics.*

*In general, this study is well designed and the methodology and sensitivity tests follow generally accepted protocols for dynamic vegetation modeling experiments. The manuscript is well written and easy to follow. The presentation could be improved, in particular the figures, and I do have a few scientific comments that should be addressed, but overall this manuscript should be suitable for publication in Biogeosciences after moderate revision.*

**Reply:** Thank you for these encouraging words and good suggestions for improvements to the manuscript. Below follow a few discussion points to your questions and point-to-point answers, as well as an updated manuscript with your suggestions incorporated or discussed.

**General comments**
*I would like to see more discussion of the processes that were not included in the model, concentrating on the following:*

*1. Seed dispersal limitations to vegetation change: A great deal of modeling work has gone in to understanding the role of seed dispersal in limiting plant migration rates, particularly along vertical gradients similar to the principle one at Abisko. The pioneer in this research has been Heike Lischke with her TreeMig Model (Lischke et al., 2006), which has been applied to the Arctic (Epstein et al., 2007), and some representation of dispersal and migration has even been incorporated into a version of LPJ developed in part by your colleagues in Lund (Lehsten et al., 2019). Of course, dispersal*

*limitations are not the whole story of what limits plant migration (Scherrer et al., 2020). Nevertheless, some further discussion on this topic and additional citations would be welcome in this manuscript, as it is effectively missing at the moment.*

**Reply:**  Thank you for a good comment! As you state, a great deal of research has gone into the study of this topic, where the TreeMIG-model stands out as a pioneering example. Seed dispersal limitations were only briefly touched upon in our Introduction, where we state that latitudinal treelines might be more sensitive to dispersal constraints than elevational treelines (see L49-50), as has been discussed by Rees et al. (2020). LPJ-GUESS assumes the presence of seeds in all grid cells, meaning that simulated species or PFTs can establish once the climate is favourable, as defined by each individual's predefined bioclimatic limits (Smith et al. 2014).  While seed dispersal limitations might play a role in constraining larger scale, latitudinal, migration rates, we do not believe that it will play any major role in our case, across distances of the order of tens of metres. This is evidenced by the presence of tree saplings above the current treeline in the Abisko area (Sundqvist et al., 2008) and that some of these stunted mountain birch individuals can be a few decades old (Hofgaard et al., 2009). The latter suggesting that these seedlings have been able to spread and establish above the treeline in historic times, but not develop into mature trees. There should thus not be any constraint on the availability of the birch seeds in the seedbank above the treeline (Truong et al., 2007). According to your suggestion we added a section in the discussion (see L509-521) about this topic.

"LPJ-GUESS assumes the presence of seeds in all gridcells and PFTs may establish when the 20-year (running) average climate is within PFT-specific bioclimatic limits for establishment. This assumption may overlook potential constraints on plant migration rates such as seed dispersal and reproduction. On larger spatial scales, it is likely that lags in range shifts would arise from these additional constraints (Rees et al., 2020; Brown et al., 2018). Models that account for dispersal limitations generally predict slower latitudinal tree migration than models driven solely by climate (Epstein et al., 2007). However, on smaller spatial scales, the same models predict competitive interactions to be more dominant in determining species migration rates (Scherrer et al., 2020). In a seed transplant study from the Swiss alps, seed viability could not been shown to decline towards the range limits of eight European broadleaved tree species (Kollas et al., 2012; Körner et al., 2016). Similarly, gene flow above the treeline could not be shown to be limited to near-treeline trees in the Abisko region (Truong et al., 2007). Furthermore, tree saplings have been reported to be common up to 100m above the present treeline (Sundqvist et al., 2008; Hofgaard et al., 2009). As environmental conditions improve, these individuals may form the new treeline. Thus, on the scales considered in this study, we do not regard dispersal limitations as a major factor in limiting range shifts of trees."

*2. Permafrost effects: While it is mentioned that the study area lies in the permafrost zone, it is not really discussed how changing permafrost dynamics; deepening of the active layer, changes in effective rooting depth, changing water table depth, etc. affects vegetation. At the ultra-high resolution used in the model simulations, it might be important to account for how ground freezing and soil vertical and horizontal movement caused by frost (or lack of it in the future) could affect survival and competition among the various plant functional types simulated. While I appreciate that a full treatment of permafrost dynamics may be beyond the scope of the present study, it would be*

*good to have some further discussion/speculation of how this process could influence the results and conclusions of the modeling work performed here.*

**Reply:** Permafrost aspects are important in the Arctic. Our study area is located within a zone of discontinuous permafrost. In our study domain, the presence of permafrost is limited to wetlands, e.g., the Stordalen area, or at the highest elevations (Callaghan et al., 2013). As we do not include any analysis of wetlands in our study, the only permafrost zones in our study domain are located on the mountains above treeline. Thus, permafrost and treelines in our area are far separated and permafrost effects on the treeline or shrub growth will be of minor importance for our results. Permafrost dynamics may be important for vegetation dynamics at the highest elevations in our historic simulations.

The model simulates the effect of frozen ground and permafrost (where present) on water availability, though the effects of frost horizontal and vertical movement are not taken into account. Such processes could affect survival and competition among the plant functional types, especially in the seedling stage when plants are most vulnerable to mechanical disturbance. These effects could be relevant to treeline dynamics at the high grid resolution of our study, but are not accounted for by our model. We have added a few lines about permafrost dynamics in the results (L287-290; 363-365) and discussion (L 488-495).

L287-290
"Simulated permafrost with an active layer thickness of <1.5 m was present at elevations down to 560 m a.s.l. in a few gridcells, but was always well above the treeline. More shallow permafrost (<1 m) was only present in gridcells at elevations of 940 m a.s.l. and above."

L 363-365
"Permafrost with an active layer thickness of <1.5m disappeared completely from our study domain in all scenarios except the coldest (GFDL-ESM2M-RCP2.6) where it occurred in a few gridcells at elevations of approximately 600 m a.s.l. However, the shallow permafrost (<1m) had disappeared also in this scenario."

L 488-495

"Permafrost was only present at the highest elevations during the historic simulation but had disappeared from the landscape at the end of the century for all except the coldest scenario (GFDL-ESM2M-RCP2.6). The simulated permafrost was however always well above the treeline and did not have a significant impact on the treeline advancement. While some aspects of ground freezing are accounted for in the model, soil vertical and horizontal movement caused by frost, and amelioration of such effects in the warmer future climate are not. Such processes could affect survival and competition among the plant functional types, especially in the seedling stage when plants are most vulnerable to mechanical disturbance (Holtmeier and Broll, 2007). These effects could be relevant to treeline dynamics at the high grid resolution of our study but are not accounted for by our model."

*3. Slope and aspect effects: It is mentioned that the study area is hilly or even mountainous; at the resolution of the model, how were slope and aspect handled? Particularly in a high-latitude situation*

*with low sun angle, slope and aspect must be very important in influencing the surface radiation budget, soil temperature, and snowpack dynamics. If slope and aspect were not considered in this study, some explanation of why is required, and similarly to the point above, the authors should include some discussion of the potential effects that this could have on their results. Furthermore, as the resolution of the modeling approaches micrometeorological scale, it would be helpful to have some further discussion of how the lakeshore climate may be different from areas further away, e.g., with respect to wind speed and the radiative environment.*

**Reply:** The questions about the influence of slope and aspect will of course be of importance when modelling on this scale. Elevation (i.e. lapse rate) is the main driver of mesoscale temperature variation in our climate dataset. Effects of mountainside aspect are included in the climate dataset by Yang et al. (2011) as local variations to the temperature data alongside the lake effect by Lake Torneträsk. Both surface temperature and radiation are input variables to the model. Any relationships between surface temperature and radiation are not calculated within the model but are implicitly included in the dataset. We have therefore included a statement of these properties of the climate dataset in the methods section (see L 165-172). For a full description of the climate dataset we refer to Yang et al. (2011; 2012). Soil temperature and snowpack dynamics are determined by the pre-calculated surface air temperature and precipitation inputs, and soil properties, with no explicit use of aspect or slope in the model's process descriptions. Effects of wind is not regarded in the model and we do not use wind as input to any parts of the model.

Line 165-172
"The field measurements were conducted in form of transects that captured mesoscale climatic variations, i.e., lapse rates. In addition, the transects were placed to capture microclimatic effects of the nearby lake Torneträsk and variations in radiation stemming from mountainside aspect. The temperature in the lower parts of the Abisko valley in the resulting dataset was influenced by the lake with milder winters and less yearly variability. At higher elevation, the temperature was more variable over the year and the local scale variations were more dependent on the different solar angles between seasons and mountainside aspect (Yang et al., 2011; Yang et al., 2012)"

*4. Linked to the point above on slope and aspect, I would have liked to see some more discussion of the spatial heterogeneity of the snowpack. Again, at the model resolution and over the spatial domain considered, I would imagine that the formation of snowdrifts and other snowpack variation is important for soil temperature and moisture, plant survival, and N cycling. Numerous studies have demonstrated that wind and slope/aspect have a strong influence on the depth and density of snow in snowdrifts and on the rate and timing of snowmelt. This spatial heterogeneity in snowpack depth and melt rate affects winter surface temperatures and therefore survival of plants at and above the treeline, and growing season soil moisture (there are many studies on this topic but one example is Walker et al., 1999). Again, while a full treatment of snowpack heterogeneity might be beyond the scope of the study, some more discussion of this important process, and how it might influence the region around Abisko specifically, is warranted.*

**Reply:** Our version of the model does not include any formulation of snowdrift or wind compaction. The simulated spatial heterogeneity of the snowpack in the model is thus minimal (<1%) for the winter (DJF) months. Snow trapping processes or any potential snow-shrub feedbacks (Sturm et al., 2001) are therefore not included in our simulations. These processes are

undoubtedly important for soil temperatures and subsequently mineralisation of soil organic matter. The model accounts for snow insulation effects on soil temperature, but do not have any representation of frost damage, which might affect seedling survival and be ameliorated by an insulating snow cover. As the model does not account for heterogeneity over the landscape in the compaction and drifting of snow, this might lead to an overestimation of winter soil temperatures above the treeline, where high winds and low roughness tend to deplete snow cover, and an underestimation of winter soil temperatures in the forest, where the opposite is true. We have extended the discussion to include these feedbacks (see L 475-483).

Line 475-483

"Furthermore, our model does not include any wind related processes such as wind mediated snow transport or compaction. Thus, our simulations result in a homogenous snowpack during the winter months with no differentiation in sheltering or frost damage that may result from different snow and ice properties. Sheltered locations in the landscape are known to promote survival of tree saplings (Sundqvist et al., 2008). For N cycling this may also mean that suggested snow-shrub feedbacks (Sturm et al., 2001; Sturm, 2005) are not possible to capture with the current version of our model. While overall rates of advance were captured, local variations arising from physical barriers such as steep slopes, stony patches or anthropogenic disturbances were consequently not possible to capture as these processes are not implemented in the model."

*5. Given the overall importance of N cycling for the results of this study, it would be helpful to have an overview of the N module in LPJ-GUESS. In particular, I would like to understand how biological N fixation is represented and if certain PFTs (e.g., something representing Alnus spp.) can be advantaged in nitrogen poor settings because they are capable of enhanced N fixation especially with warmer temperatures.*

**Reply:**   A more detailed description of the N cycle in LPJ-GUESS have been included in the manuscript (see L 126-129). Full details and equations, which are too extensive to repeat in the present paper, are presented in the cited paper by Smith et al. (2014).

Line 126-129

"Biological N fixation is represented by an empirical relationship between annual evapotranspiration and nitrogen fixation (Cleveland et al., 1999) and occurs differently within each patch. Additional inputs of nitrogen to the system occur through nitrogen deposition or fertilisation. Nitrogen is lost from the system through leaching, gaseous emissions from soils and wildfires. For a full description of the nitrogen cycle in LPJ-GUESS, see Smith et al. (2014)."

**Specific comments**
*Line 146*
*It is mentioned that three replicate patches in each gridcell are used for LPJ-GUESS. It is worth going in to a little more detail here to justify this choice of the number of replicate patches. As I understand, each patch in LPJ-GUESS is meant to represent an area of 0.1 ha. With a 50m grid (cells of 0.25 ha), three replicates effectively makes an explicit representation of the entire gridcell, no?*

**Reply:** The replicate patches are intended to give an estimation of landscape-scale heterogeneity within a gridcell or stand that might arise from spatial variation in stochastic processes and histories in the model (e.g. stochastic establishment, mortality and patch destroying disturbance events). No assumptions are made about how the patches are distributed within a wider area, they are merely a statistical sample of equally possible disturbance/demographic histories across the landscape of a grid cell. We have adjusted the text to improve clarity on lines 148-149.

"No assumptions are made about how the patches are distributed within a gridcell, they are a statistical sample of equally possible disturbance/demographic histories across the landscape of a gridcell."

*Line 163*
*Further to my general comment above, please explain how slope and aspect are incorporated into this gridded climatology.*

**Reply:** We have added a clarification of how the dataset was constructed. In general, the field measurements were set up in forms of transects with temperature loggers. These transects were selected to cover variations in mesoscale climate patterns that arise from elevation (i.e., lapse rate), and local variations (aspect and lake effects). See lines 165-172 and quoted text above under point 3.

*Line 169 and Fig S1.1*
*From looking at the figure I don't really see how temperature is "more variable" with increase in elevation. Perhaps some descriptive statistics would be more useful here*

**Reply:** We acknowledge that the statement about climate variability is not evident from the figure. It is however a feature of the climate dataset described closer in Yang et al. (2012). We have updated the figure with larger elevational bands for clarity and values written out in the figure. In addition, we have added a panel with the standard deviation for each elevational band and month. This panel may be interpreted as the magnitude of the local effect at each elevational band and month.

*Line 178-183*
*Soil edaphic controls on vegetation are an important part of treeline and subarctic biogeography; it is even mentioned on this line. So why not make any attempt to account for spatial variation in soils? Although the spatial resolution is still a bit coarse, why not use the pedometrics-based Soilgrids250 (Hengl et al., 2017) instead of simply prescribing the same soils everywhere? Could you have done some sensitivity tests to quantify the model response of vegetation distribution and treeline to different soil physical properties?*

**Reply:** We agree that soil factors are undoubtedly important also for soil nutrient cycling and storage. We assumed a uniform soil texture within our study domain sourced from our standard soil dataset (Batjes, 2005) as input to the model. We came to this decision after we had contacted the Swedish Geological Survey (SGU), which we judged as the best source of more detailed soil data or surveys. A more detailed survey of soil texture for this area was however non-existent. The dataset by Hengl et al. (2017)is indeed impressive and

includes a few processes known to create variations in soil texture over the landscape. However, we would argue that a dataset with higher resolution is not necessarily a more reliable or accurate dataset. Furthermore, the variations in soil textures within our study domain are not larger than approximately 3% for the clay or sand fractions. Such small variations in soil texture would not generate any significant changes to the landscape heterogeneity. We did some sensitivity tests in a sub-section of our domain during the preparation phase of the simulations. While some factors such as soil organic carbon content was affected, more drastic variations in soil texture are needed to affect vegetation distribution or treeline dynamics to any large degree. We have not included these sensitivity tests in our results. We have added a sentence in the discussion (L483-486) about this limitation in soil texture heterogeneity in our model input.

Line 483-486

"High-resolution, local observations of vertically-resolved soil texture and soil organic matter content (see, e.g. Hengl et al. (2017) for an example compiled using machine-learning) have the potential to improve the spatial variability of modelled soil temperatures and nutrient cycling in our study domain. We will investigate this uncertainty in future studies."

Line 473

Where are these transects? Call out the supplementary figure here or even better refer to an overview map (see comment below). How were the locations and orientations of these transects chosen?

**Reply:** The figure has been referenced in the text on line 239. We also include a short sentence in the methods section (see L 240-243) and figure caption (supplementary materials) about how these transects were selected.

Line 240-243

These transects were chosen to represent a large spread in heterogeneity with regard to slope and aspect in the landscape. A subsample of the selected transects were placed close to the transects used by Van Bogaert et al. (2011) and used to evaluate model performance. Results from the model evaluation are summarised in Table 1 and Table S1.1

**Comments on presentation**

*I would appreciate seeing an overview map or aerial photo of the study area showing topography and the location of the lake, any rivers, and settlements, roads, etc. I would also like to see at least an inset map showing the location of the study area within Europe and Sweden.*

**Reply:** We have provided an inset map of Sweden and the location of Abisko in figure 2. In this figure we added contour lines to mark the landscape topography. We also marked the lake and the location of the Abisko scientific station (ANS) in this figure. We thank the reviewer for these suggestions which we believe improves the maps a lot. We do not include settlements, roads and rivers as our simulations are not affected by these features.

*Fig. 2*

*What is the white area in these maps? Why are the colors used for the PFTs in Fig. 2a not the same as those used in Fig. 3? Please harmonize. Please add a scale bar to these figures, and perhaps one or two longitude and latitude tick marks/labels. As many readers look only at the figures, or the figures first, it would be helpful to spell out the PFT names in the figure legend here instead of making the reader refer back to an additional table or text to decode these.*

**Reply:**    We have updated the figure and figure caption in accordance with these suggestions.

*Fig. 3*
*Harmonize the colors with Fig. 2a*

**Reply:**    We have updated the figure with harmonized colors as in figure 2.

*Fig. S1.1*
*The figure caption appears to be cut off*

**Reply:**    Fixed.

*Supplement S1 Table 1*
*What is reported in the column "Reported (van Bogart et al. 2011)"? What are these units of?*

 **Reply:**    These are estimated treeline migration in elevational meters reported by van Bogart et al (2011). We have updated this table to clarify this.

*Fig. S1.4*
*What is the gray scape in plotted in the background of the map? What is the white area?*

 **Reply:**    The gray scape in this figure is the landscape relief and the white area is Lake Torneträsk. We have clarified this in the figure caption.

*Fig. S1.6e*
*What is the principle control on annual shortwave radiation? Is it cloud cover? This could also be discussed in the main text.*

**Reply:**    The principal control of the annual shortwave radiation within the global climate models (GCMs) is cloud cover. However, we use the monthly bias adjusted shortwave radiation-output from the GCMs as input to our model.  The annual values are averages of the monthly output provided by the GCMs. We have updated the figure caption to clarify this.

**Reply to reviewer 2**

*Gustafson et al. aim at modelling future treeline position in northern Sweden and the causes that control potential shifts. I am familiar with the associated theory and the region. Using environmental data and biological response functions, a digital vegetation model is applied (with the treeline forming species of this region, Betula pubescence ssp.).*

**Reply:** We would like to thank Professor Körner for his comments on our manuscript and for giving us the opportunity to clarify a few aspects of our narrative.

*Such a model has a predefined response hierarchy, that is, assumptions on both, the relative importance of drivers and the direction of causality. These assumptions, though absolutely central, are not mentioned upfront, but they become obvious as one reads the text.*

**Reply:** All models are simplifications of reality and have assumptions underpinning the processes and parameters included in the model. One of the motivations of this study and the use of such a high-resolution grid was to test some of the assumptions built into the model. The scale used in the study enables comparison to experimental and observational data, which is more difficult to do when, as is typical, the model output represents the average over a large spatial scale. The Abisko region has been thoroughly studied with diverse observational datasets, providing an ideal laboratory for benchmarking a model. A modelling study complements empirical approaches in interpreting the causalities in observed dynamics of treeline advance and vegetation shifts. Lastly, the study aims to simulate future vegetation shifts in the area, something only a model can do.

*One of the key assumptions is that these trees are C limited and that photosynthesis, A f (T, PFD, CO2) drives growth. Starting with such assumptions, the inevitable outcome is that $CO_2$ matters for growth, although it may not matter for treeline position, depending in other assumptions. Yet, in my view this is dressing the horse from the tail. It became obvious in recent decades that outside horticulture and agro-conditions, growth controls A via phloem downloading on demand for C, and this demand is set by meristem activity and other sinks. Not surprisingly manipulating C supply in the field neither rose growth or productivity in alpine vegetation, nor in treeline trees (there were transitory effects on young, isolated Larix individuals in exceptionally warm summers, that did not affect final biomass data, pine was never affected). None of these works are cited (the only reference to $CO_2$ experiments is the differential response of two upper montane understorey shrubs).*

*I do respect the skills of the authors to parameterize and handle such a complex set of algorithms, but the underlaying rationale reflects our understanding of causalities in the 1980s. I am quite aware that starting with modelling growth rises other issues, but several teams have no engaged (Simone Fatichi, Andrew Friend, several papers) and even the dendro-community is now moving forward in that direction (read e.g. Jan Tumajer et al. Frontiers in Plant Sciences 28 Jan 2021). They are still unable to handle resource supply as modulating factors.*

**Reply:** LPJ-GUESS includes representations of several processes that can interact to drive or constrain changes in vegetation composition and ecosystem functioning, and moreover these vary over time in response to the evolving ecosystem state. An example is the treeline and productivity changes investigated in our study. Specifically, the following are tested:

- Climatic – We simulate the influence of historic and future climate (temperature, precipitation and solar radiation) change through a range of scenarios extracted from the CMIP5 project (Taylor et al., 2012).
- Nutrient limitation – This is tested in different nitrogen deposition scenarios (see section 2.4.3 in the manuscript) where we vary the nitrogen load in our projection simulations.
- Productivity – We mainly test the influence of productivity on vegetation shifts through our $CO_2$-fertilisation experiment (see section 2.4.2 in the manuscript).

- Ecological – A prerequisite for treeline advance is the establishment and growth of trees above the current treeline. Tree seedlings will have to compete with existing vegetation during their establishment (Grau et al., 2012; Lett and Dorrepaal, 2018). Our model includes a representation of interspecific competition for light, soil water and nutrients.

Productivity does not drive treeline advance – we agree with this, in fact it is an outcome of our study, demonstrating that emergent dynamics can not necessarily be trivially predicted from the inputs and modelled processes. Specifically, in our $CO_2$-only experiment, the treeline did not advance despite increased GPP (Photosynthesis) (See Sec. 3.3.2 & Fig. 5b). If anything, this indicates that the lack of a correlation between tree productivity and treeline advance in the real world could have more than one mechanistic interpretation. There is no scientific consensus, yet that meristem activity universally explains treeline position. Indeed, it would be strange if evolution were not 'smart' enough to make maximal use of available resources to drive fitness.

Our simulation results are not inconsistent with the observation of ample carbon storage in trees close to the treeline (Hoch and Körner, 2012), which indicates that trees close to the treelines do not suffer from carbon shortage. To further highlight this, we have added results for NPP to the results (L266-269; L301-306) and discussion (L405-407; L440-453) sections of the manuscript. We do acknowledge (Table 1) that the simulated over-estimation of biomass is a limitation of the model and an area where potential improvements can be made. While a few factors (e.g., herbivory from both mammals and insects, mainly *Epirrita autumnata*) may reconcile the model results with the observations, they cannot fully explain our over-estimation of biomass. Recognising this, and in line with the comments by the reviewer, we put forward temperature limitations on xylogenesis (wood formation) as a potential area for model improvements in the future, as has also been done by others (e.g., Friend et al., 2019; Leuzinger et al., 2013; Pugh et al., 2016) - see our Discussion (see L456-464). This will be of importance for future projections of boreal and Arctic carbon budget estimations and could potentially alter the simulated treeline advance in our Abisko domain. Lastly, our modelled treeline advance is not only constrained by tree physiological factors (e.g., xylogenesis or photosynthesis), but also by ecological factors (e.g., interspecific plant competition and soil nutrient cycling). The importance of including ecological processes in model studies of treeline rather than solely considering bioclimatic limits to treeline advance has also been emphasised by others (e.g., Scherrer et al., 2020). The model does however include hard limits to vegetation distribution through the bioclimatic envelope of each PFT. We have added a few sentences in the methods to clarify this (see L154-157).

We note that more mechanistic treatments of migration and the spread of seeds could also play a role, as could altered disturbance patterns, type and intensity, as was also pointed out by reviewer #1.

Added texts in the manuscript:

Line 154-157:

[revised manuscript text omitted]

*With these concerns, the results of the modelling reflects the assumptions. If one assumes soil fertility matters for treeline trees and selects N to represent these nutrients, the outcome is that N matters.*

**Reply:** Dynamic vegetation models (DVMs) are designed to test multiple and interactive driving factors and processes under different environmental conditions. LPJ-GUESS (Smith et al., 2001; Smith et al., 2014), does not build on the assumption of any single driving or modulating factor. The importance of each driver is an emergent outcome of the simulated dynamics in response to variation in the drivers, and the evolving state of the system (e.g. soil N availability, plant community structure) which also modify the processes. Thus we respectfully disagree with the statement "*the results of the modelling reflects the assumptions*". While this is true by definition for any model, the point the reviewer is making, that the simulated treeline dynamics could be trivially predicted, knowing the input to the model and the process formulations used, is not correct.

*If soil fertility were controlling treeline position there should not be a global treeline isotherm and treeline should be at higher elevation on good soils and at lower elevations on poor soils, not what we see in the field (e.g. soils developing on young glacial deposits versus treelines on geologically old, weathered, low latitude mountains.*

**Reply:** The reviewer's comment states that, given our conclusions, local variations in soil fertility would give rise to differences in treeline elevation and that no such pattern is seen in global treeline records. This would be true if no recognition of temperature as a controlling factor were made. We do not dispute the importance of temperature as an overall control on treeline position, but stand by our conclusion, consistent with physiological, demographic, and ecological principles and assumptions around which there is broad scientific consensus, that nitrogen limitation will constrain rates of future treeline advance in this area.

The global correlation between treelines and the 6.7°C isotherm (6-7°C in arctic/boreal regions; Körner and Paulsen, 2004) is not disputed in this study. In fact, results from our historic simulations corroborate this, at least for this region since the position of the simulated treeline correlates well with an isotherm close to this limit (Fig S1.4; supplementary materials). The lag arises in our future projections of treeline advance when the isotherm displacement is more rapid than the treeline advance in most scenarios (Fig 4). We traced this effect to N-limitation in the soils as we see a faster treeline migration in our climate change only simulations compared to the future projections when N-deposition is decreasing.

*Experiments by Hoch (2013) revealed that there is no compensatory effect of nutrient addition to low temperature constraints of growth. There is also no direct link between tree vigor and treeline position. Trees at treelines in Bolivia and Tibet at close to 5000 m elevation hardly grow (minute tree ring width), because they are clearly moisture and thus, nutrient limited.*

**Reply:** The model simulates induced treeline advance when only nitrogen deposition increased (Fig 5c). This was a result of an increased allocation to above-ground biomass with a subsequent

advantage in the light competition. The reviewer refers to the study by Hoch (2012) as evidence against such an effect. However, the Hoch study was performed in a fully factorial design with seedlings grown in phytotrons with controlled environment at 6 °C or 12 °C and half of the seedlings fertilised. The cold temperature seedlings showed no response to added fertiliser, while warm temperature seedlings showed a strong biomass increase with fertilisation. We would argue that such a controlled climate study may not mimic the role of nitrogen effects in a full ecosystem setting.

*I hope these comments are useful for revisiting the rationales underpinned in this model. I read the other report in copernicus, thanks for providing it. It seems to address additional facets of treeline formation, but does not touch upon the more fundamental bias regarding the assumptions that drive the model output.*

**Reply:** We would like again to thank Professor Körner for his comments. We would also argue that our study makes a meaningful contribution to the treeline literature, and hope that our findings will stimulate further experimental and modelling studies of treeline advance and associated feedbacks to climate.

**References**

[revised manuscript text omitted]

---

## Author Response (AR2)

**Editor comment**

The revised manuscript was re-read by both reviewers, who as before have thoughtful, probing comments and suggestions. Both referees appreciate that you have put a significant amount of effort into revisions, and responded effectively to most of the previous comments. Both recommend only minor revisions. That said, both focus on a common theme: it has to be very clear, in both the title and manuscript, (i) the fact that this is modeling study, (ii) the input variables required to run the model (cf. R2), and (iii) how the model inputs and logic are linked to the conclusions. R1 put this issue succinctly: "I do think readers have a right to read the authors' explanation of how the modelling framework employed (the hierarchy of drivers in particular) led to the main conclusion."

In summary, then, this is a strong manuscript and the reviewers (and I) appreciate your thoughtful responses and revisions made. There remain some minor revisions, mostly related to methodological clarity and being careful about the scope of the inferences as conveyed in the conclusions and title.

**Reply:**

We would like to thank the editor for his encouraging words and for facilitating the review process. We have answered all the reviewers' comments below. The updated manuscript has a more extensive description of the processes represented in the model. The title is also updated to clarify that this is a modelling study. Furthermore, we have updated the text where any confusion regarding the methodology could arise. Lastly, we have gone through the text to correct spelling errors, grammatical errors and improved clarity.

**Reviewer 1 – Christian Körner**

*This revised version pertains some of the points I had suggested to be adjusted. While valuing such a modelling attempt, it seems imperative that one makes a clear distinction between empirical facts and suggestions derived from a model. The title reads as if N controls treeline in this region, while this is, at best, a suggestion the authors deduce from a model that they configurated in a way that N plays a key role. I accept the point that all models need to assume certain input variables and a certain hierarchy of causalities. I had hoped the authors write a brief paragraph in which they explain the main assumptions that drive this model. These could include that they assume that C is a limiting resource for growth (hence a high priority in the chain of causalities), and because photosynthesis (A) requires N, this is causing growth to be N controlled etc.). Once there are a few sentences of this kind, every reader understands the rationale behind this model and can make up his/her own mind about the results.*

**Reply:** We would like to once again thank Professor Körner for his thoughtful comments on our manuscript.

We have updated the methods section to further clarify how the model represents photosynthesis and growth, see lines 122-132 Furthermore, we have updated the title to clarify that this is a modelling study. However, we do reiterate our early point that the importance of each driving or modulating factor is an emergent outcome of the model and that the model has not been configured to assume nitrogen to be limiting.

L122-132

Canopy fluxes of carbon dioxide and water vapour are calculated by a coupled photosynthesis and stomatal conductance scheme based on the approach of BIOME3 (Haxeltine and Prentice, 1996). Photosynthesis is a function of air temperature, incoming shortwave or photosynthetically active radiation, $[CO_2]$, and water and nutrient availability. Autotrophic respiration has three components - maintenance, growth, and leaf respiration. Tissue maintenance respiration is dependent on soil and air temperature for root and above-ground respiration, respectively, along with a dependency on tissue C:N stoichiometry. All assimilated carbon that is not consumed by autotrophic respiration, less a 10% flux to reproductive organs, is allocated to leaves, fine roots and, for woody PFTs, sapwood, following a set of prescribed allometric relationships for each PFT, resulting in biomass, height and diameter growth (Sitch et al., 2003). Consequently, an individual in the model is assumed to be carbon limited when autotrophic respiration equals or exceeds the amount of carbon assimilated by photosynthesis. Chronically negative carbon balance at the individual level eventually results in plant death.

*Since the results contradict what we know from observational data about treeline formation, the discussion would best explain why the authors think the model is still correct. By observational data, I mean the evidence that the treeline position tracks an isotherm globally, irrespective of local soil fertility. One could for instance argue that this correlation has an error term and within that variance, there is space for a nutrient effect. The authors could further argue why the site for which they model future treeline position is special in terms of tree nutrition, thus opening space for what the model outcome suggests. I do think readers have a right to read the authors' explanation of how the modelling framework employed (the hierarchy of drivers in particular) led to the main conclusion. I think this is within the tradition of modelling that model results are discussed with reference empirical data (which can be critisized as well, of course).*

**Reply:** As stated earlier, we do not disagree with the evidence of a global correlation between treeline position and an isotherm of 6-7 ℃. We also make no claim that treelines should be at higher elevation on more fertile soils. What the study shows is that it is *plausible* in a rapidly changing environment with multiple environmental drivers and lagged ecological and biogeochemical responses for the historically observed correlation between treeline position and temperature to be broken. In our model, this arises as a consequence of nutrient shortage and competition with already established vegetation. The evidence for the global correlation between treelines and an isotherm is strong, but it is by necessity a snapshot in time. As such, it is not necessarily a strong predictor for future treelines. Here we want to highlight the distinction between current or future treeline position and the rate of treeline advancement. We have added a few sentences about this in the discussion, as an attempt to clarify these differences (see lines 427-431; lines 446-448).

We agree that modelling studies should evaluate the model's performance against available evidence. The paper discusses both long-term (e.g. treeline positions inferred from pollen records; lines 454-462) and short-term (rates of advance during the last 100 years; lines 452-453 + Table 1) treeline positions and rates of advance. Drivers or modulators for treeline advance that are discussed include the carbon limitation hypothesis (lines 474-487), permafrost effects (lines 520-528), physical barriers (lines 505-513), competitive effects on vegetation dynamics (lines 541-552) and nutrient cycling (lines 425-451).

Lines 427-431

During our historic simulations, the treeline correlated well with a soil temperature isotherm close to the globally observed 6-7°C isotherm. However, in our projection period the correlation between the treeline position and the isotherm weakened, revealing a fading or potential lag of the treeline-climate equilibrium that became stronger with increased warming. Future rates of treeline advance were thus constrained by factors other than temperature in our simulations.

Lines 446-448
Historically, treeline positions show a strong correlation with the 6-7°C isotherm (Körner and Paulsen, 2004). These records are, however, a snapshot in time and are not necessarily a strong predictor of future treeline, with other factors (as for nitrogen in our results) potentially breaking the link to temperature.

*I am aware of the common jargon, but still find it needs more caution when the word 'test' is employed. We tested N deposition..., we tested CO2 effects... would better read: we performed test runs with the model to explore its sensitivity to...*

**Reply:** We have updated the text and believe we are expressing ourselves accurately in the few instances where we use the word "tests".

*On meristems: these are just one of the components it needs to build a tree. Mycorrhiza is another one. LPG assumes A drives meristems. Fine, why not say so? Others found it is the other way round. This is merely an issue of open debate and the need of an attempt to justify why the authors think A drives meristems (sensu growth).*

**Reply**: It is not photosynthesis (A) but NPP that drives growth in LPJ-GUESS, where it is assumed that all NPP except for a 10% deduction for reproduction drives plant growth. We are aware that this assumption is subject to discussion, and we discuss this on lines 474-487. We have also clarified this assumption on lines 130-131.

Lines 130-131
… Consequently, an individual in the model is assumed to be carbon limited when autotrophic respiration equals or exceeds the amount of carbon assimilated by photosynthesis …

*On plant competition: sure this matters, in the seedling stage. So if there are good reasons to assume a recruitment limitation, a few words would be welcome on whether the model assumes a dominance of symmetric (soil-root) or asymmetric (light, LAI) competition, and for which life stage such interactions are assumed to be critical. A seed limitation can be excluded in mountain birch, given the billions of light weight seeds produced ever autumn. What matters is (a) the seedling establishment process within the aerodynamic boundary layer close to the ground and (b) the emergence of saplings from that layer. These are two entirely different steps in establishing a young tree.*

**Reply:** We have revised the model description section in methods substantially, and now describe in some detail how the model represents these processes, see lines 135-138. We have also added detail in the Discussion to clarify how various assumptions influence the simulated treeline dynamics (see lines 440-442)

Lines 135-138

Competition for light and nutrients is assumed to be asymmetric, i.e., individuals with taller canopies or larger root systems will be advantaged in the capture of resources under scarcity. Water uptake is divided equally among individuals according to the water availability and the fraction of each PFT's roots occupying each soil layer.

Lines 440-442
For treeline advance to occur, trees need to invade the space already occupied by other vegetation. As the model assumes asymmetric competition for nutrients, newly established seedlings have a disadvantage compared to incumbent vegetation, further slowing down the modelled rate of treeline advance.

*Bioclimatic envelope: it would be good to read which thermal envelope the model assumes if any. I mean, all organisms have a thermal limit somewhere and mountain birch is no exception. Whatever envelope the model employs and how this envelope is assumed to act upon trees should be explained (it could be an influence on A, on respiration, on meristems, on freezing tolerance, on microbial soil activity... or a combination of all; we simply need to be told, how the range limit is defined in this LPG variant).*

**Reply:** We have added information in the model description section in methods to describe how the bioclimatic envelope sets the hard limits to PFT ranges in the model (see lines 174-175). Specific PFT parameters can be found in Supplementary Table S2.2.

Lines 174-175
The parameters are intended to capture broader climatic properties of each gridcell. A detailed description of each bioclimatic parameter and its respective values can be found in Supplementary Table S2.2

*In summary, this papers requires more explicit statements on the drivers of tree responses assumed. It cannot be left to the reader to find this out in the LPG script.*

**Reply:** We appreciate the thoughtful comments from Professor Körner. We hope that the additional model detail and interpretation in the manuscript, as outlined above, make it clear and understandable for readers.

Reviewer 2 – Anonymous

*Review of*

*Nitrogen restricts future treeline advance in the sub-arctic*

*By Adrian Gustafson, et al.*

*In their revised manuscript, the authors have generally done a good job of responding to the detailed and critical reviewer comments.*

*I still have a few open issues with the manuscript, that should be able to be addressed with minor revisions.*

*Line numbers refer to the clean copy of the revised manuscript.*

**Reply:** We would like to thank the reviewer for the thoughtful comments and for once again taking the time to scrutinise our paper.

*Line 126*
*So, biological nitrogen fixation does not depend on the plant species/functional type assemblage that is present? That is to say, there are no plant types/species that are more likely to be associated with biological N fixation than others? I guess not in this model, but it would be good to add that precision here.*

**Reply:** This assumption is correct; no PFT has intrinsically greater biological N fixation ability in the model. We have clarified this on lines 143-145. In addition, very few species in the study area form root association with N-fixing microbes and those *Alnus spp* and *Fabaceae* species that can be found are never dominant in the local plant communities.

Lines 143-145
LPJ-GUESS does not currently incorporate PFT-specific nitrogen fixation, which for instance may be associated with species that form root nodules, such as *Alnu*s spp.

*Lines 139-146*
*Could you please put the content of this into a table? I find it quite irritating to constantly be looking back in the text to try to find the acronyms describing the plant functional type and species names. The authors did not accept my suggestion to spell out the plant names in the color bar on Figure 2, but they did put the abbreviations into the caption. Nevertheless, it would be much easier to read if there was a simple table in to accompany the text with the plant name and abbreviation.*

**Reply:** The information about each PFT and their target species is provided in Supplementary Table S2.1. Plant functional types do not necessarily represent a single species, but rather a group of species with similar functional and physiognomic traits. However, a PFT may have a target species which the modeller had in mind when parameterising the PFT. In the case of IBS, mountain birch (*Betula pubescens ssp. czerepanovii*) which forms the treeline in our study region was the target species. This PFT may also include species such as rowan (*Sorbus aucuparia),* which is also a species growing in the forest-tundra ecotone in the area, but of less significance.

*Line 159*
*There appears to have been some misunderstanding of my comment about the replicate patches. While the authors now provide some additional information on the meaning of the replicate patch, they still do not explain/justify how/why they chose to use three patches per gridcell. Why not four patches, or 10, or two? Please add a simple explanation here.*

**Reply:** We apologise for the misunderstanding. We have added a motivation for the 3 patches on lines 178-179.

Lines 178-179
We judged this number sufficient to obtain a stable representation of vegetation dynamics given the relative area of each gridcell and replicate patches (0.1 ha).

*Line 162*
*Just below the section header here, please briefly summarize all of the input variables required to run*

*LPJ-GUESS. This information is too difficult to find as it is scattered throughout the following paragraphs. Picking through the text I could infer that LPJ-GUESS is driven by monthly mean temperature, precipitation, and solar radiation, and soil texture. Is solar radiation downwelling shortwave only? In what units? What about longwave fluxes? What about slope and aspect? Minimum and maximum temperature or diurnal temperature range, or only the mean? How many soil layers, and to what depth? Coarse fragments or only texture fractions? Alternatively, you could provide a table with this information.*

**Reply**: We have added information about inputs to the requested section, see lines 184-187. However, for clarity we only include variables that are used within the model, and thus exclude variables that are not used, e.g., long wave radiation fluxes, slope and aspect, etc. We also included a sentence in the model description about the number of soil layers, see line 150.

Line 150
The soil scheme includes 15 equally distributed soil layers constituting a total soil depth of 1.5 meters.

Lines 184-187
The input variables used as forcing in LPJ-GUESS simulations are monthly 2m air temperature (°C), precipitation (mm), and incoming shortwave radiation (W m$^{-2}$) as well as annual atmospheric [$CO_2$] (ppm), soil texture (mineral fractions only), and nitrogen deposition (kg N ha$^{-1}$ month$^{-1}$). Monthly air temperature and shortwave radiation are interpolated to a daily time-step while precipitation is randomly distributed over the month using monthly wet-days.

*Line 174*
*I also infer here that a wet-day probability is also required input to LPJ-GUESS. How was CRUNCEP downscaled to the appropriate resolution? How was the wet-day probability handled for the future climate runs? Please add wet day probability to the information above.*

**Reply:** The reviewer's assumption is correct. The wet-day probability is used to randomly distribute monthly precipitation over the days in the month. This input was retrieved from the CRUNCEP dataset and used in its 'raw' form. The observational precipitation dataset from Abisko had a monthly resolution and did thus not provide any additional information for adjusting any bias in the number of wet days per months. For our projection simulations we assumed the historic 1971-2000 wet-day climatology to be constant also in the future. As the number of wet days per month was relatively high (often >20 days per month), we do not believe that this assumption had a significant influence on our results. We added a clarification of this on lines 218-219.

Lines 218-219
The number of wet-days per month was assumed not to change in the future scenario simulations, so we used the 1971-2000 climatology for this period.

*Figure 2*
*Figure 2 is much improved but I cannot see the location of the Abisko station marked on the map even though this is promised in the response to reviewers. Also, why have you made*

*these maps so tiny? Biogeosciences is not a tabloid journal, and you should have nothing to hide. I recommend increasing the size of each of the four maps in this figure by 100%. These are key results and deserve to be easier to scrutinize. Also, could you please add a scale bar in meters or km?*

**Reply**: We have added the Abisko station and a scale bar to the map. Furthermore, we have increased the size of the plots.

*Figure 3*
*Could you please put the color bar in the same order as the way it is presented in Figure 2? Yes, I know this is a little thing, but irritating nevertheless.*

**Reply:** Done

*Figure S1.4*
*You could plot Lake Tornetrask on these maps in blue as it is done in Figure 2. Much easier to read.*

**Reply:** Done